# Development and implementation of a Digital Twin workshop for a smoke alarm production line

Min Wu[1]*, Wenfeng Ying[2]

1 School of Mechanical and Electrical Engineering, Ningbo Polytechnic University, Ningbo, Zhejiang, China, 2 Ningbo Weizmat Electronics Co., Ltd., Ningbo, Zhejiang, China

* 53760012@qq.com

## Abstract

To address issues such as low manual production efficiency, inability to track production status, and lack of traceability for quality defects in smoke alarm manufacturing, this paper proposes a method for constructing a digital twin workshop for smoke alarm production lines. First, through demand analysis of the smoke alarm production workshop, a digital twin workshop system architecture is established. Then, by constructing a workshop digital twin model, building the workshop information system architecture, and completing multi-source heterogeneous data collection, a virtual-physical mapping between the virtual workshop and the physical workshop is achieved. Finally, the feasibility is validated in a company's smoke alarm production workshop, demonstrating that the system can realize functions such as production progress tracking, quality control, and equipment operation monitoring. Compared to traditional transmission production, the application of this digital twin workshop significantly enhances production efficiency. Defect identification rates increased by 3.2%, product defect rates dropped significantly below 0.1%, and labor costs decreased by 96.7%.

## 1. Introduction

As manufacturing undergoes a profound transformation toward high-end, intelligent, and green development, digital twins—as the core enabling technology for digital manufacturing—have emerged as a pivotal force in solving complex manufacturing system control challenges and advancing the implementation of smart manufacturing. This is achieved through their core advantages of virtual-physical mapping, real-time interaction, and end-to-end process optimization. Digital twins utilize virtualization technology to digitize physical industrial entities, constructing precisely corresponding digital models that enable simulation analysis, process optimization, predictive decision-making, and full lifecycle management. Their technical characteristics align exceptionally well with the manufacturing industry's demands for high-mix, fast-paced, and high-quality production.

**Data availability statement:** All data generated or analyzed during this study are included in this published article.

**Funding:** The author(s) received no specific funding for this work.

**Competing interests:** The authors have declared that no competing interests exist.

Currently, digital twin technology has been extensively explored and validated across multiple manufacturing domains.Tao, F. et al. [1] systematically investigated the theoretical foundations and practical implementation pathways for constructing digital twin workshop models.Ferreira, F. et al. [2] applied it to classic car restoration, optimizing workshop operational efficiency through virtual simulation. Dinh, M.-C. et al. [3] developed fault diagnosis algorithms for wind turbines based on a digital twin framework, enabling virtual monitoring of equipment status.Schabany, D. et al. [4] constructed a maturity assessment model for battery factories, providing standardized references for industrial applications.Neugebauer, J. et al. [5] optimized container terminal scheduling processes through digital twins. Barni, A. et al. [6] utilized discrete event simulation to address production line cycle time fluctuations. Sujova, E. et al. [7] established a twin model for engine block machining lines to optimize process parameters.Cohen, Y. et al. [8] employed twin technology for real-time tracking and progress control of workpieces on assembly lines. These studies span diverse scenarios including discrete and process manufacturing, validating the universal value of digital twins in managing complex systems.

Meanwhile, numerous studies have highlighted the unique advantages of digital twins by comparing them with traditional manufacturing technologies. The data-driven twin modeling approach proposed by Semeraro, C. et al. [9] demonstrates superior flexibility and scalability compared to traditional rule-based modeling in complex production systems. Pronost, G. et al. [10] pointed out the limitations of traditional simulation techniques in real-time performance and cross-scale integration, while digital twins enable dynamic optimization throughout the product lifecycle, reducing R&D cycles by 28%. Mayer, F. et al. [11] emphasized the advantages of digital twins in multiphysics coupling modeling and interdisciplinary collaboration, improving the modeling accuracy of complex systems by 42%. Fu, X. et al. [12] validated that a twin-driven quality characterization framework achieves 51% higher accuracy than traditional offline inspection in real-time error compensation; Alfaro, VD. et al. [13] demonstrated the dynamic decision-making advantages of twin technology in predictive maintenance, reducing equipment downtime by 34%; Corsini, RR. et al. [14] constructed a digital twin supply chain model demonstrating significantly superior robustness under dynamic disturbance.s compared to traditional static planning, reducing logistics costs by 29%; Lee, D. et al. [15] developed a production line design optimization system that reduced reconfiguration time by 40%. These studies collectively demonstrate that digital twins compensate for the shortcomings of traditional manufacturing technologies in dynamic response and complex system management by offering advantages in cycle time reduction, cost control, and real-time optimization.

As a critical security product safeguarding lives and property, smoke alarms exhibit production characteristics and quality requirements that make them highly compatible with digital twin technology. From a production perspective, smoke alarms encompass diverse categories (including photoelectric, ionization, and hybrid types) and must adapt to varied application scenarios (residential, industrial, public buildings,etc.), This results in multifaceted production tasks and frequent line changes. industry research indicates an average changeover time of 2.3 hours, significantly

higher than in large-scale manufacturing sectors like automotive components [16]. Concurrently, annual production volumes per factory typically exceed 5 million units, demanding production cycles under 12 seconds per unit and placing stringent requirements on process continuity and efficiency stability. From a quality perspective, core metrics like smoke alarm detection sensitivity (0.65dB/m~1.0dB/m) and alarm response time (≤30 seconds) directly impact safety assurance. This necessitates comprehensive quality traceability and precise control throughout the entire production process,a requirement that aligns perfectly with the core capabilities of digital twins: bidirectional mapping between virtual and physical systems and integrated data across all elements. Research confirms that digital twins can reduce changeover time for multi-product lines by over 40% through virtual simulation, elevate precision assembly defect detection rates to 99.7% via real-time data interaction, and satisfy full-process data traceability requirements for UL 217 (U.S.), GB 20517 (China), and other mandatory safety certifications for security products. This establishes digital twins as the optimal technical solution for overcoming production challenges in smoke alarm manufacturing.

Compared to mature manufacturing sectors such as automobiles, engines, and batteries, the smoke alarm industry faces distinct challenges in applying digital twins, specifically manifested in three aspects:

The issue of weak data foundations is even more pronounced: 62% of the industry's production equipment consists of outdated machinery with a service life exceeding eight years. There is a lack of unified data interfaces such as OPC UA and MTConnect, with only 38% of equipment capable of automatic data collection—in contrast, the automotive manufacturing sector typically achieves over 90% equipment data collection rates. Additionally, manual operations account for 65% of processes (primarily sensor calibration, circuit board soldering, etc.), leading to production data being predominantly recorded on paper and entered manually. This results in severe data fragmentation, making the integration of heterogeneous data far more challenging than in highly automated manufacturing sectors.

Multi-scale modeling demands stricter precision requirements: Smoke alarm products typically measure 80 mm × 80 mm × 50 mm, housing internal components such as sensor chips (≤5 mm × 5 mm), a buzzer (≤20 mm diameter), and other precision components. The manufacturing process involves precision soldering with 0.3 mm pitch solder joints and component assembly tolerances controlled to ±0.02 mm. This necessitates cross-scale precision modeling that simultaneously addresses micro-processes (solder joint formation, sensor calibration) and macro-production lines (equipment load, material flow). — modeling errors must be controlled within 3%, whereas the tolerance for modeling errors in large-scale mechanical manufacturing is typically 5%–8%.

Stricter safety and compliance requirements: Products must pass mandatory certifications in multiple countries, while production processes must comply with standards such as ISO 9001:2015 and IEC 61508. Digital twin models not only need to meet production optimization needs but also must undergo virtual validation to proactively mitigate compliance risks. For instance, models must simulate production processes under varying temperature and humidity conditions (−10°C to 55°C) to ensure product qualification rates remain consistently above 99.5%. Failure to do so risks certification invalidation, extending model development and validation cycles by 30% compared to conventional manufacturing.

Currently, smoke alarm production still faces significant challenges: On one hand, the manual-dominated production management model struggles to accommodate diverse product lines and high-volume production demands, resulting in an efficiency rate of only 32 units per worker per hour—58% lower than digital production lines—with a human error rate reaching 1.8%. On the other hand, the absence of real-time collection and visual display of production equipment and quality data prevents tracing 83% of quality defects to specific processes or operators, creating significant safety risks. Compounded by unique industry challenges, these issues further underscore the urgency of digital transformation.

Based on this, this paper proposes a digital workshop model based on digital twins, using a smoke alarm production line as the research subject. By establishing precise bidirectional mapping and real-time interaction between the physical workshop and virtual workshop, it achieves comprehensive integration of all elements, processes, and operational data. Driven by twin data, it facilitates iterative optimization of production factor management, production planning, and production process control across the physical workshop, virtual workshop, and service systems, ultimately enhancing both

production efficiency and quality control. This research specifically addresses the unique challenges of applying digital twins in the smoke alarm industry, laying the foundation for its transition from traditional manufacturing to digital transformation. It also provides a practical paradigm for the digital upgrade of precision security product manufacturing.

## 2. Physical system composition and process flow of the smoke alarm production line

### 2.1. Physical system composition

The production line for smoke alarm comprises five core subsystems (as depicted in Fig 1). The equipment and functions of each subsystem are detailed below:

Injection Molding Subsystem: this includes one horizontal injection molding machine (with a clamping force of 500 KN), a raw material dryer, and a cooling water circulation system. It is responsible for melting, injecting, pressure-holding, and cooling the plastic raw materials to form the base.

Assembly Subsystem: comprising two six-axis robots (with a repeat positioning accuracy of ±0.02 mm), a conveyor belt (operating at a speed of 0.5 m/s), and tooling fixtures, this subsystem handles the assembly of the base and metal inserts.

Machine Vision Subsystem: this subsystem includes two industrial cameras (each with 5 million pixels), lenses (with a focal length of 16 mm), an annular LED light source, and an image processing unit. It is used for assembly positioning guidance and quality inspection.

Detection Subsystem: equipped with size detection sensors (with an accuracy of ±0.01 mm) and appearance defect detection devices, this subsystem conducts a comprehensive inspection of the assembled bases.

Packing and Sealing Subsystem: this subsystem includes an automatic packing machine, a tape sealing machine, and a labeling machine, which together facilitate the packaging and identification of qualified bases.

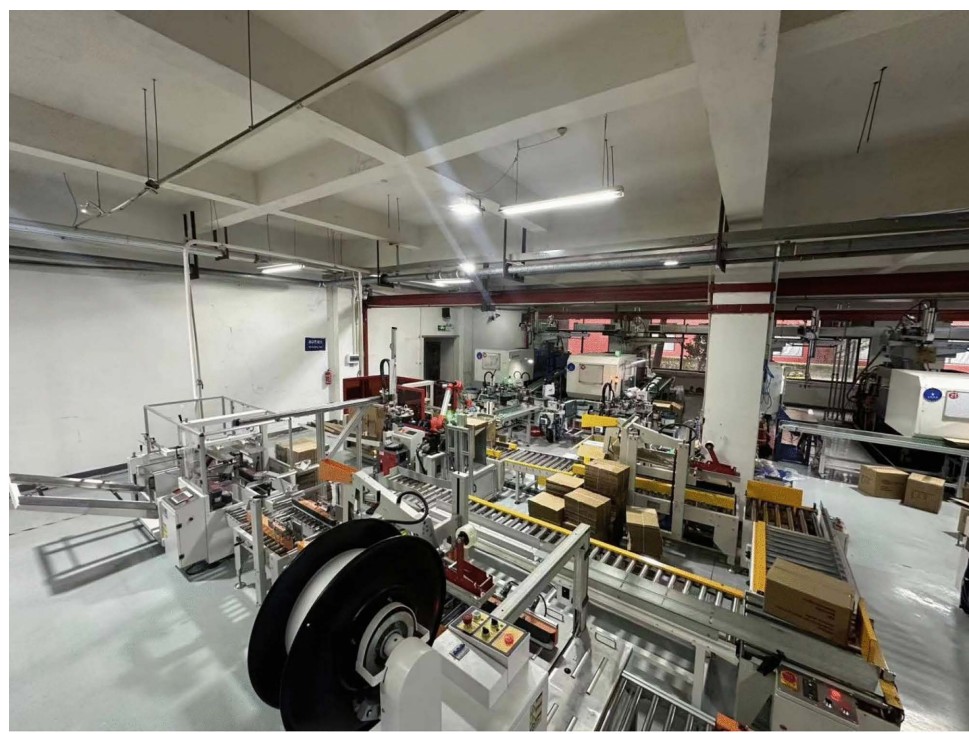

**Fig 1. Physical drawing of workshop.**

## 2.2. Process flow analysis

The entire production process of the base comprises four critical stages, each with distinct process parameters and sequential logic.

Injection Molding Stage: Raw materials are first dried at 80°C for 4 hours and then fed into the barrel of the injection molding machine. Heated to a nozzle temperature of 220°C, the materials melt and are injected into the mold cavity under a pressure of 80 MPa, which is held for 3 seconds. After a 15-second cooling period, the mold opens, and the base blank is ejected.

Manipulator Assembly Stage: A conveyor transports the base blank to the assembly station. Here, a machine vision system captures images of the positioning holes on the base. After calculating positional deviations, the system guides a manipulator to grasp a metal insert (8 mm in diameter) and press it into the predefined location on the base with a force of 500 N.

Visual Inspection Stage: At the inspection station, an industrial camera captures 360-degree images of the assembled base. Using image processing algorithms (e.g., template matching and edge detection), the system evaluates the perpendicularity of the inserted assembly (tolerance: ± 0.1 mm) and identifies any surface defects, such as cracks or material shortages.

Packing and Sealing Stage: Qualified bases are conveyed to the packing machine, where they are stacked into cartons in batches of 20. A sealing machine then applies tape to the top and bottom of each carton. Finally, a labeling machine prints and affixes a QR code label containing the production date and batch information.

The entire process is coordinated by a programmable logic controller(PLC), with state feedback provided by sensors (e.g., proximity switches and pressure sensors), forming a seamless and automated production line.

## 3. Architecture of digital twin workshop system

With reference to the system architecture of a digital twin workshop [17], the digital twin workshop for the smoke alarm production line consists of four layers: the physical layer, the model layer, the data layer, and the service layer, as illustrated in Fig 2.

The physical layer forms the foundation of the digital twin workshop. It comprises workshop entities such as personnel, machinery, and materials, along with all production activities carried out in the physical workshop. Key components include production and packaging machinery, assembly lines, inspection and transmission systems, storage facilities, sensors, PLCs, RFID tags, and other devices responsible for data acquisition and communication. This layer provides the essential data required for modeling and virtual simulation.

At the core of the digital twin workshop, the model layer represents the virtual workshop and its associated production processes. It includes various models, rules, and knowledge bases, as well as functionalities for simulation, analysis, optimization, and decision-making. The virtual workshop serves as a real-time digital counterpart of the physical workshop.

The data layer acts as a service platform that provides data support to the physical, model, and service layers. It integrates equipment, material, environmental, personnel, and production process data, enabling interconnectivity and data mapping across all layers.

Finally, the service layer embodies the application level of the digital twin workshop, enabling workshop visualization and data-driven functionalities. It offers services such as real-time alerts, production process management and control, and equipment management.

## 4. Construction of workshop digital twin model

The twin model of the workshop production process serves as the foundation for constructing a digital twin workshop, providing an accurate digital representation of the physical workshop. Key elements of the production process include personnel, machinery, materials, and the environment.

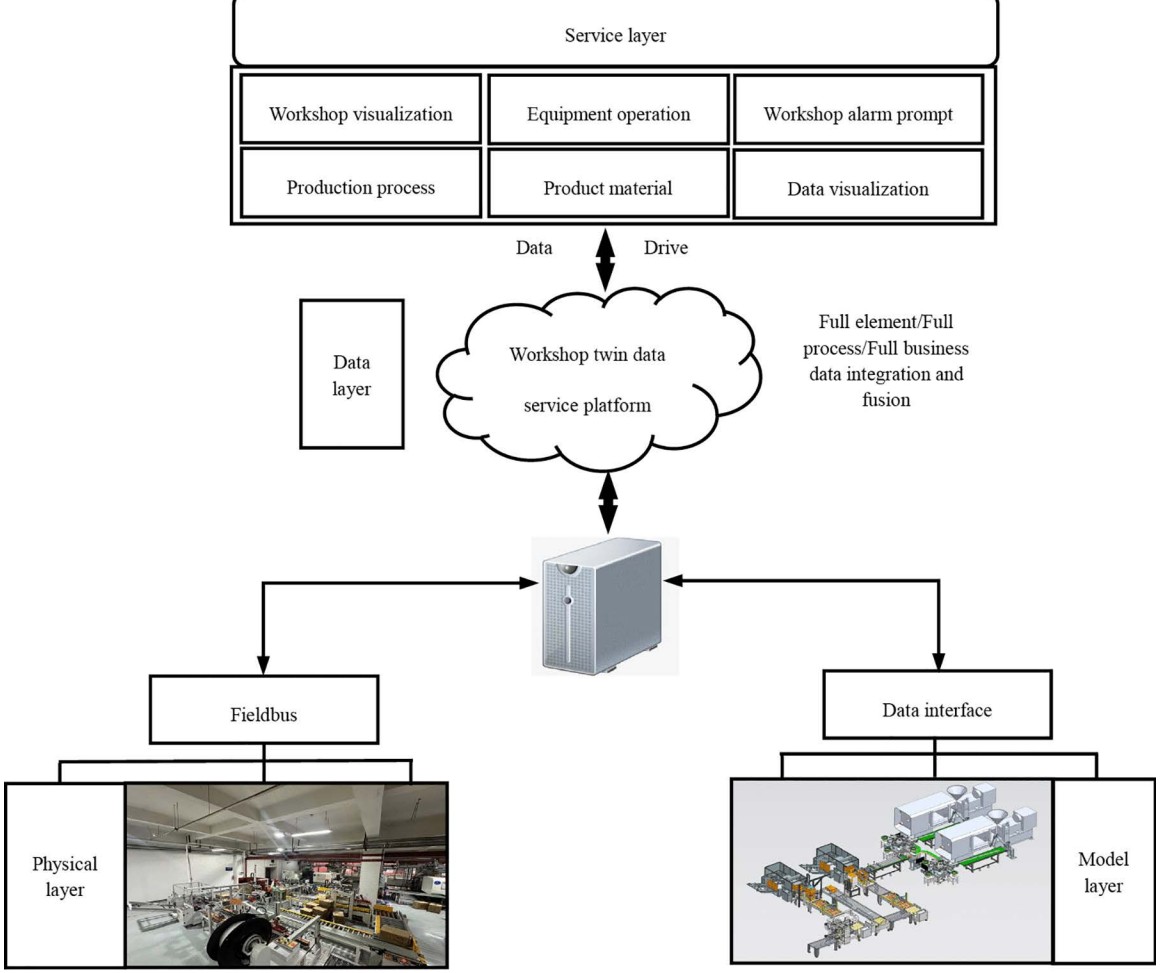

**Fig 2. Architecture of digital twin workshop for smoke alarm production line.**

Based on the structure and workflow of the smoke alarm production line, the workshop twin model is categorized into three levels: unit, production line, and workshop. The unit level comprises equipment twin models, personnel models, environmental models, and product/material models. At the production line level, integration and assembly are achieved by incorporating spatial relationships, constraint relationships, and other associative connections among units. The workshop level enables interconnection, interoperability, and collaborative optimization across different systems through information flow, energy flow, and other means, building upon the production line level.

To construct a digital twin model of the workshop (as shown in Fig 3), the following steps are taken. First, the dimensions, shape, assembly relationships, and other attributes and rules of the physical entities are mapped, allowing the model to accurately represent the geometric and physical properties of the smoke alarm production workshop based on known parameters. Second, the behavior of physical entities is represented within the model, and multi-disciplinary, multi-physical-quantity, and multi-scale data—such as mechanical, electrical, process control, and functional description data—are acquired through communication interfaces. Finally, an information model is established that gathers large-scale data from both physical and model layers via data acquisition and virtual simulation. By comparing and analyzing

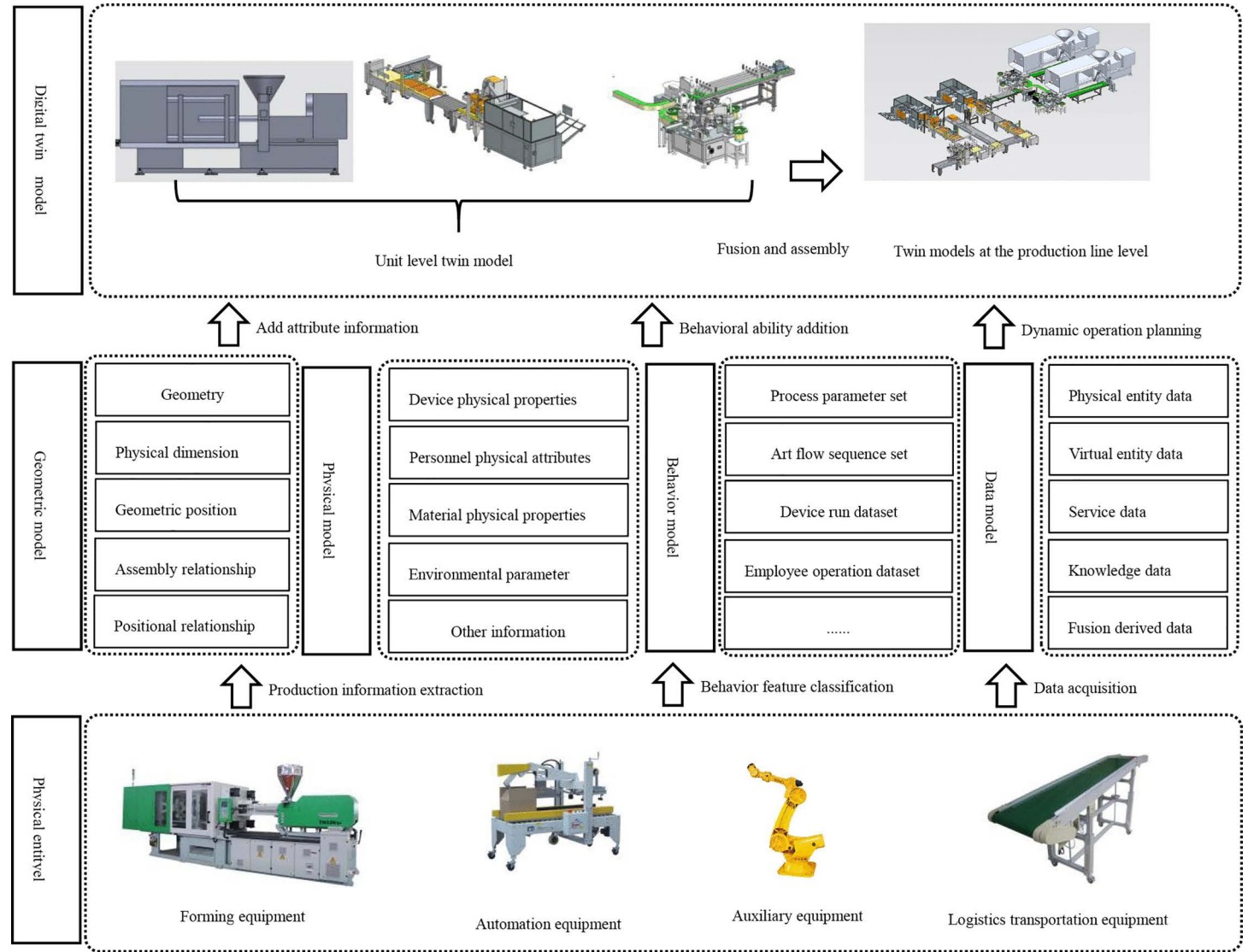

**Fig 3. Modeling process of twin workshop for smoke alarm production line.**

data between the physical and virtual workshops, decisions are made according to parameter relationships and rules, and feedback is provided to guide the physical workshop, enabling optimized control of the entire production process.

Thus, each model integrates foundational models from multiple disciplines—such as mechanical, electronic, and hydraulic systems—as well as multi-dimensional representations of geometry, physics, behavior, and rules. The mathematical expression for the twin model of the smoke alarm workshop is given in Equation (1).

$$WTM_{sa} = \{G_{sa}, P_{sa}, B_{sa}, DI\} \tag{1}$$

In the formula, $WTM_{sa}$ is the twin model of the smoke alarm production workshop, $G_{sa}$ is the geometric model, $P_{sa}$ is the physical model, $Bsa$ is the behavioral model, and $DI$ is the data information model.

## 4.1. Geometric model

The geometric model provides a three-dimensional representation of a physical entity's shape, size, position, and assembly relationships, maintaining strong spatiotemporal consistency with the real-world object. A tree-structured hierarchy is established based on the structural relationships within the smoke alarm production workshop, as shown in Fig 4. Using the workshop as the root node, the model is divided into four modules: equipment, personnel, material, and environment. Each node is assigned a local coordinate system to store its positional coordinates relative to its parent node.

In Siemens NX software, each submodule component is meticulously drafted at a 1:1 scale and assembled based on parent-child relationships to form individual submodule models. These are then arranged according to the actual workshop layout to create a preliminary workshop model. To improve computational efficiency in graphics processing and simulation, the 3D models undergo lightweight processing by eliminating unnecessary lines and surfaces. For components with minimal impact on simulation accuracy, only overall dimensions and motion patterns are represented, omitting ancillary structures. The mathematical formulation of the geometric model is given in Equation (2).

$$G_{sa} = \{S_g, D_g, P_g, AR_g, PR_g\} \tag{2}$$

In the formula: $S_g$ is the geometric shape, $D_g$ is the geometric dimension, $P_g$ is the geometric position, $AR_g$ is the assembly relationship, and $PR_g$ is the positional relationship [11].

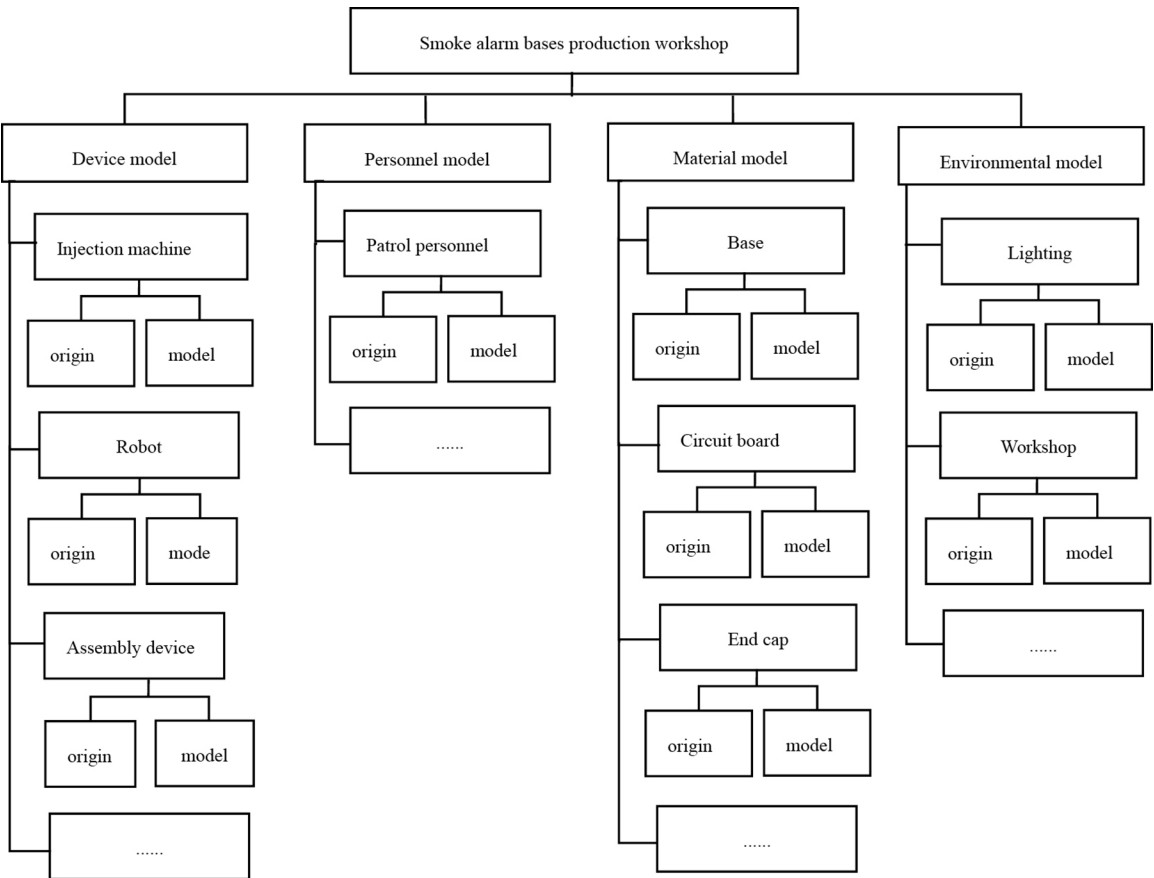

**Fig 4. Tree diagram of smoke alarm production workshop model.**

## 4.2. Physical model

The primary role of a physical model is to represent the physical properties of equipment, materials, and personnel, including mechanical, thermal, and optical characteristics of devices such as injection molding machines, machine vision systems, and robotic arms. By incorporating attributes such as material type, mass, friction, and collision bodies into the geometric model, the physical behavior of the equipment is accurately simulated. Furthermore, the model integrates various types of sensors to capture real-time data on material flow and environmental conditions. The mathematical representation of this physical model is provided as Equation (3).

$$P_{sa} = \{E_p, M_p, P_p PC, EP, \text{Info}\} \tag{3}$$

In the formula, $E_p$ is the physical attribute of the equipment, $M_p$ is the physical attribute of the material, $P_p$ is the physical attribute of the personnel, $PC$ is the physical characteristic of the equipment, $EP$ is the environmental parameter, and Info is other information.

The physical attributes of equipment mainly include type number, element name, physical parameters, and production and processing status.Among them, the type number is the identification of all devices. Element names are used to distinguish between different digital twin models of the same manufacturing equipment abstract model.The physical parameters mainly include transmission speed, cutting force, maximum capacity and other data, and the physical parameters of different devices are different.The production and processing status refers to the state at different times during the operation of the equipment. The mathematical expression is (4).

$$E_p = \{TM, Name, PP, MSS\} \tag{4}$$

In the formula, $TM$ is the device type number, $Name$ is the feature name, $PP$ is the physical parameter of the device, and $MSS$ is the model state set.

The physical properties of materials mainly include material name, material variety, material model, material status, etc. The mathematical expression is (5):

$$M_p = \{M_{name}, M_v, M_t, M_s...\} \tag{5}$$

In the formula, $M_{name}$ is the material name, $M_v$ is the material variety, $M_t$ is the material model, and $M_s$ is the material status.

The physical attributes of personnel mainly include personnel number, personnel name, personnel occupation, personnel shift, etc. The mathematical expression is (6):

$$P_p = \{P_i, P_n, P_w, P_c...\} \tag{6}$$

In the formula, $Pi$ personnel number, $P_n$ personnel name, $Pw$ personnel job type, $P_c$ personnel shift.

The physical properties of the environment include temperature, humidity, illumination, air pressure, etc. The mathematical expression is (7):

$$EP = \{EP_{tem}, EP_{hum}, EP_l, EP_p...\} \tag{7}$$

In the formula, $EP_{tem}$ is the ambient temperature, $EP_{hum}$ is the ambient humidity, $EP_l$ is the illumination, $EP_p$ is the air pressure.

## 4.3. Behavioral model

The behavior model describes the production processes of the smoke alarm, including the injection molding of shells from raw materials within a specified timeframe, internal operational mechanisms, component assembly, quality inspection, and packaging. Behavior models are classified into sensor modeling and actuator modeling [18].

Equipment operational parameters are configured according to specified requirements, with corresponding sensors on the production line—such as temperature, humidity, pressure, displacement, flow, speed, position, and vision sensors—defined based on their working principles. Through sensor behavior modeling, signals from the digital twin model are fed back to the control system based on operational status.

To improve transition accuracy between process stages and reduce errors caused solely by computational triggers, position sensors are installed at the entry and exit of each stage. Equipment operation timing is determined by a combination of operational condition calculations, trigger signals from outlet sensors by preceding products, and inlet sensor triggers by subsequent products. This method enhances production accuracy and success rates.

Based on the working principles that drive the equipment, motion control parameters of the corresponding actuators are defined and linked to control system signals and the motion attributes of the digital model. Using actuator behavior modeling, the digital model enables the execution of mechanism movements driven by control system outputs [19].

Data collected during the production process include: process parameter set $C_{parm}$, process sequence set $P_{ss}$, equipment operation dataset $D_{data}$, part operation dataset $A_{data}$, behavior coupling relationship set $BCR$, basic event set BE, production line environment parameter set $EP$, and other relevant information. These are acquired and rendered onto the model through a data interface set. Thus, the behavior model of the smoke alarm production line can be mathematically expressed as Equation (8).

$$B_{sa} = (C_{parm}, P_{ss}, D_{data}, A_{data}, BCR, BE, EP, Info\ldots)$$

(8)

## 4.4. Construction of the digital twin based on NX

The digital twin model for the smoke alarm production line is constructed using a hierarchical approach: "equipment level → unit level → production line level." Based on the self-developed "multi-scale dynamic mapping error compensation theory".The specific framework is as follows:

Equipment Level Model: Geometric and behavioral models are established for individual equipment, including injection molding machines, manipulators, and cameras.Provide "atomic level" high-precision data support for unit level and production line level models (for example, the geometric accuracy of key components reaches 0.005 mm).

Unit Level Model: Equipment-level models are integrated into functional subsystems (e.g., injection molding, assembly) to simulate collaborative operations within each unit.Through the subsystem level dynamic error compensation algorithm, the virtual and real mapping accuracy at the subsystem level is ensured.

Production Line Level Model: Unit-level models are combined to form a virtual representation of the entire production line, enabling holistic process simulation and optimization.Based on reinforcement learning and other global optimization algorithms, the process parameters of multi subsystem coupling link are intelligently optimized.

The modeling process must adhere to three core principles"geometric consistency, behavioral equivalence, and parametric correlation"and incorporate a multi-scale dynamic mapping error compensation theory to ensure that the virtual model accurately reflects the morphological, kinematic, and performance dynamics of the physical entity under varying production loads.

### 4.4.1. Equipment-level model construction.

(1) Injection Molding Machine Modeling

Geometric Modeling:The 3D skeleton model was generated in NX using "Synchronous Modeling" technology by importing the machine's 2D CAD drawings. The machine was decomposed into sub-components (e.g., clamping mechanism, injection mechanism, ejection mechanism). Detail structures were built using "Boolean Operations" and "Feature Modeling" (e.g., extrusion, revolution). The accuracy of key components (e.g., screw, mold cavity) was controlled within 0.005 mm,

while non-critical components (e.g., housing) were simplified (removing chamfers, small holes) to reduce model complexity.At the same time, the deformation of key components under different loads is pre calculated based on finite element analysis (FEA), and the "load deformation" database is constructed to provide the basis for subsequent dynamic geometric adjustment.

Behavioral Modeling:Define the motion pair in NX MCD (clamping mechanism: rotating pair+translating pair; injection mechanism: screw pair); Through the "servo motor" to drive the motion pair, set the mold opening and closing speed (0.5m/s), injection pressure (0–100Mpa) and other driving parameters corresponding to the physical PLC control parameters.

Add "position sensor" (to detect the opening and closing state of the mold) and "pressure sensor" (to simulate the cavity pressure). The sensor signal is not only bound to the virtual model state variable, but also used as the input of the LSTM model to predict the trend of the equipment state in real time (such as warning of abnormal pressure 10 s in advance).

(2) Manipulator and Machine Vision Modeling

Manipulator Geometric Modeling:A "top-down" design approach is adopted, beginning with the construction of the manipulator and joint framework, followed by the addition of the arm, wrist, and end-effector through "assembly constraints" (coincident, parallel). The end-effector (gripper) employs a parametric design, with its opening size linked via "expressions" to accommodate inserts with diameters ranging from 5 to 10 mm. Lightweight AI models are deployed on edge computing nodes to adjust gripper parameters in real-time based on the weight of the grasped object, thereby avoiding mechanical stress overload.

Manipulator Behavioral Modeling:"Revolute Joints" were added at each joint, with rotation ranges (−180° to 180°) and angular velocities (50°/s) defined. Motion trajectories for picking, moving, and placing were predefined using NX's "Path Planning" function, utilizing arc transitions to avoid abrupt stops.Machine Vision Signal Integration: A virtual camera model was mounted on the wrist, with a field of view (60°) and resolution (5 MP) configured. The output "positioning deviation value" is quickly processed by the edge AI algorithm (improved mobilenet) and used as the intelligent compensation parameter of the manipulator movement to improve the positioning accuracy from ± 0.1 mm to ± 0.05 mm.

Machine Vision System Modeling:The geometric models of the camera, lens, and light source are built in NX, with their installation positions defined using a "coordinate system" that aligns with the physical camera's installation coordinates. Through the NX Open API, OpenCV and a lightweight AI algorithm (improved template matching combined with defect detection) are invoked to perform preliminary image processing (such as initial defect screening) at the edge layer. Only key results—such as defect coordinates and confidence levels—are uploaded to the cloud. This approach reduces the detection delay from 100ms to 30ms.

(3) Packing and Sealing Equipment Modeling

Geometric Modeling:The model of the sealing machine was structured around three key components: the conveyor rollers, the pressing roller, and the tape-cutting mechanism. The thin-shell structures (1 mm thick) were created within the "Sheet Metal Design" module. Furthermore, the carton was modeled as a flexible body using a folding design methodology, and its deformation was simulated through nonlinear finite element analysis to precisely capture its real-world behavioral changes—such as folding during the pressing operation—throughout the packaging process.

Behavioral Modeling:A "Revolute Joint" was added to the conveyor roller table, with its speed set to 100 RPM.A "Prismatic Joint" was added to the pressing roller mechanism, with a stroke (0–50 mm) defined and a pressure parameter (100 N) linked to a virtual pressure sensor. The tape-cutting action was simulated via "Boolean Operations," with a cutting delay time set to match the physical solenoid valve's response.The pressure data of the pressing roller is synchronously input into the AI prediction model to predict the carton pressing quality (such as whether there are indentation defects).

**4.4.2. Production line-level model integration.** Layout Integration:In the NX "Assembly" environment, all equipment models were positioned according to the physical workshop layout, aligning the coordinate origin with the workshop's ground reference point."Distance Constraints" and "Angle Constraints" ensured relative positional accuracy (±5 mm); for example, the gap between the injection molding machine and the conveyor belt was set to 10 mm.At the same time, based on the digital twin lightweight algorithm (deleting invisible parts and simplifying non critical surfaces), the total number of triangular patches of the model is controlled within 5 million, taking into account the geometric accuracy and real-time rendering efficiency.

Logical Integration:The NX MCD "Signal Flow" module was used to define inter-device control signals (e.g., an ejection signal from the injection molding machine triggers conveyor startup, while an assembly completion signal triggers camera capture at the inspection station). A hybrid control architecture, "edge logic + cloud-based decision-making," was implemented: the edge layer (comprising industrial gateways) processes time-critical signals (e.g., emergency stops, sensor anomalies), whereas the cloud executes global production scheduling via a virtual PLC using ladder logic.

Performance Optimization:Beyond lightweight data processing, a "Layered Display + Edge Rendering" strategy was adopted. Depending on the simulation viewpoint, edge nodes handle local model rendering (e.g., details of a single device), while the cloud handles global scene rendering (e.g., the overall line layout). This method maintains a real-time rendering frame rate above 30 fps, meeting the requirements for interactive simulation.

The resulting output is presented in Fig 5.

## 5. Data acquisition and transmission

### 5.1. Workshop information system architecture

Modeling and data interaction constitute the core of the digital twin. A well-designed cyber-physical fusion system is therefore essential for the digital twin workshop, as it enables virtual-real interactions and breaks down information silos within the digital environment.

The smoke alarm production workshop involves numerous field devices that operate on diverse communication protocols. A unified data model is critical to streamline data transmission and access. To achieve this, OPC Unified Architecture (OPC UA) servers were deployed on intelligent devices throughout the production line. Data collected from underlying

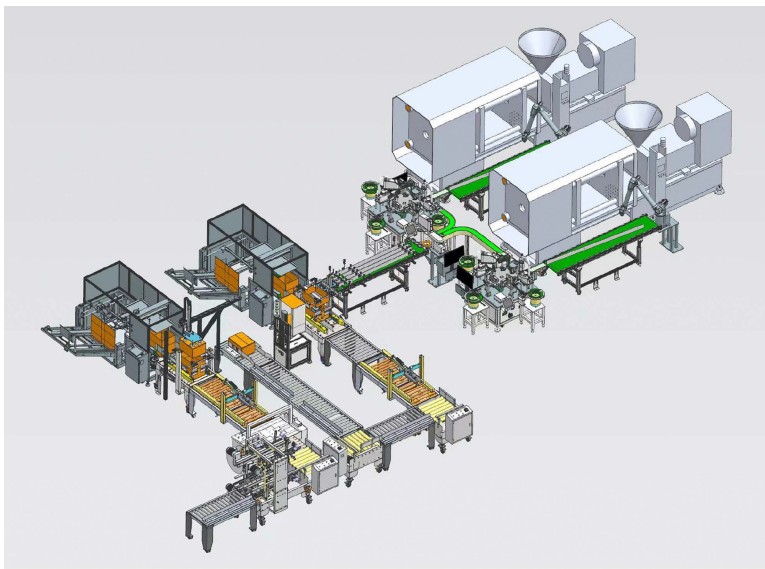

**Fig 5. Workshop digital twin model.**

equipment is processed via information modules and mapped to nodes within the OPC UA server's address space. High-volume data interactions are supported through endpoint connections to these OPC UA servers.

The OPC UA network architecture comprises three layers: the equipment layer, the communication layer, and the information layer. The equipment layer consists of production machinery, auxiliary equipment, and sensors. Operational and status data from these devices are encapsulated by the OPC UA server and transmitted to upper-level applications using either publish-subscribe or query modes. Devices may communicate directly with the OPC UA server via OPC UA clients.

The communication layer serves as a bridge connecting field devices to information systems, external networks, and cloud applications. At the information layer, an OPC UA client collects diverse data from the equipment layer and stores it in a local database. This data is then utilized by enterprise resource planning (ERP) and manufacturing execution systems (MES). The overall information system architecture of the workshop is illustrated in Fig 6.

## 5.2. Data acquisition system design

### 5.2.1. Types and parameters of collected data.
The construction of a digital twin workshop relies on the comprehensive acquisition and deep integration of multi-source heterogeneous data from the physical workshop. The

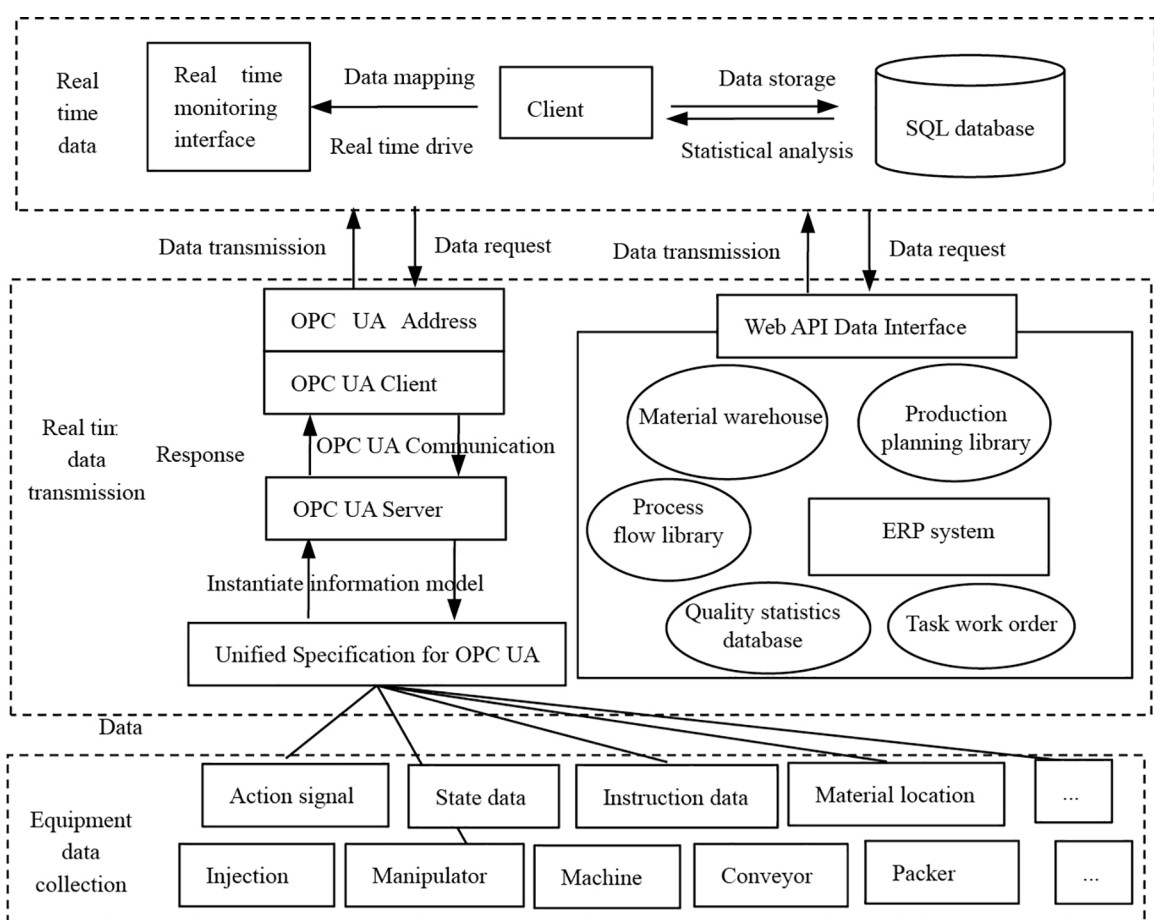

**Fig 6. The workshop's information system architecture.**

data acquisition system must cover key elements across the entire manufacturing process, including equipment status data (e.g., machine tool vibration, spindle speed, energy consumption), process parameters (e.g., processing speed, cutting depth, temperature control), product quality data (e.g., dimensional deviations, surface roughness, defect inspection results), material flow information (e.g., work-in-process (WIP) locations, inventory status, real-time delivery trajectories), and environmental parameters.

This study introduces a dual-mechanism approach—"Edge-Intelligent Acquisition + Cloud-Global Optimization"—to capture these critical elements, encompassing equipment status, process data, quality metrics, and logistics information. The optimized architecture of this system is summarized in Table 1.

**5.2.2. Hardware acquisition architecture.** Sensor Layer:A variety of sensors are installed on physical equipment. Pressure sensors (range: 0–200 MPa) and temperature sensors (PT100, −50 °C to 200 °C) are mounted on the injection molding machine.An encoder (resolution: 1024 PPR) is installed on the conveyor belt for speed monitoring.A laser displacement sensor (accuracy: ±0.001 mm) is used at the assembly station to measure insert height.

Edge Layer:Industrial gateways (e.g., Siemens IoT 2050) serve as data acquisition nodes. They connect to PLCs via PROFINET to collect equipment control signals.An OPC UA server (e.g., Kepware EX) runs on the gateway, converting sensor and PLC data into the OPC UA protocol.

Transport Layer:Industrial Ethernet (Gigabit bandwidth) forms the core communication network. Wired connections (TCP/IP) link equipment and gateways, while wireless backup (5G) is supported between gateways and servers.Network latency is controlled within 50 ms, and a priority queue mechanism ensures preferential transmission for critical data (e.g., fault signals).

## 5.3. OPC unified architecture information model construction

**5.3.1. Information model construction process.** The OPC UA based information model needs to map the physical entity and data relationship of the smoke alarm production line. The specific construction steps are as follows:

Table 1. Data acquisition parameters.

| Data Type | Collection Target | Key Parameters and Description | Collection Frequency | Accuracy Requirements |
|---|---|---|---|---|
| Equipment Status Data | Injection Molding Machine, Robot, etc. | Operation status (continuous operation/planned shutdown/fault shutdown), motor speed, temperature of key parts of equipment (such as barrel and motor housing) | 100ms high-frequency acquisition, real-time prediction of equipment status trend by edge LSTM model (early warning of potential faults 10s in advance) | The accuracy rate of state recognition is 100%; Rotational speed accuracy ±1r/min, temperature accuracy ±1 °C, combined with finite element model to dynamically correct geometric deviation. |
| Process Parameter Data | Injection Molding Machine | Injection pressure, holding time, mold temperature | 50ms acquisition, rapid fitting of parameter change curve in the edge layer, cloud reinforcement learning model to optimize the combination of process parameters | The precision of injection pressure is ±0.1MPa, the precision of holding time is ±0.01s, and the precision of mold temperature is ±1 °C. Self optimization of process parameters based on reinforcement learning |
| Quality Inspection Data | Machine Vision System | Product positioning deviation, defect type (scratch/pore/size out of tolerance, etc.), key size (diameter/length/aperture, etc.) | Each product is collected once, the edge AI (improved template matching) completes the rough detection of defects, and the cloud CNN model accurately classifies defects | The dimensional accuracy is ±0.01 mm, the accuracy of defect recognition is 99.9%, and the edge cloud collaboration improves the detection efficiency and accuracy |
| Logistics Information Data | Conveyor, AGV, Cartoning Machine, etc. | Location of logistics unit (AGV, conveyor belt carrier), conveying speed, packing quantity, three-dimensional warehouse location | 200ms acquisition, edge positioning algorithm (UWB fusion) real-time correction of location, blockchain certificate storage logistics data (tamper proof) | Position error ±5 mm, conveying speed accuracy ±0.05m/s, packing counting accuracy 100%, blockchain ensures the reliability of logistics data |

(1) Namespace definition

Define custom namespace (URI: http://base-production-line.com/opcua), distinguished from OPC UA standard namespace, Namespace contains object nodes such as "production line", "production unit" and "assembly unit".

(2) Object level design

Root node: baseproductionline.

   First level child nodes: injectionunit (production unit), assemblyunit (assembly unit), inspectionunit (inspection unit), packingunit (packaging unit).

   Secondary child nodes: each unit contains specific equipment objects (such as injectionmoldingmachine, robot, camera, etc.).

   Three level child nodes: the device object contains variables and method nodes such as status, parameters, and operation (such as the injectionpressure variable and the start method).

   Each node adds semantic tags (such as "injection molding machine - clamping mechanism - pressure") to support cross system semantic retrieval and intelligent understanding.

(3) Variable node attribute definition

Taking the "Injection Pressure" variable of an injection molding machine as an example, in addition to defining basic attributes (Data type: Float; Engineering unit: MPa; Min: 0; Max: 100; Access permission: Read/Write; Sampling interval: 50ms), intelligent attributes such as "AI-predicted value" and "Historical optimal value" are incorporated. This enables the virtual model to actively recommend process parameters based on historical data, achieving an intelligent interaction where the virtual model guides physical production.

**5.3.2. Information model of smoke alarm production line.** Based on the organizational structure, functional modules, and data flow requirements within the smoke alarm production workshop, the physical equipment and production process data are digitally described and modeled. Organized according to information modeling principles, these form the digital twin workshop information model for smoke alarm production, which includes a static attribute set, a process attribute set, and a functional component set [20].

   The workshop information model corresponds to the entire workshop. The functional component set represents each functional module within the workshop, while the resource component set denotes the resources involved in each module. The attribute set consists of a collection of attributes, which serve as the basic descriptive units within the information model. These attributes, with their fixed structure, describe the relationships among devices, components, and other attribute sets.

   The smoke alarm production line involves a wide variety of equipment types, presenting significant integration challenges. Effective equipment integration and data acquisition are crucial to the performance of the digital twin workshop. OPC Unified Architecture (OPC UA) provides a standardized communication mechanism that supports synchronous, asynchronous, and distributed communication, enabling horizontal and vertical access to diverse types of data.

   Based on OPC UA, this paper constructs an information model for the digital twin workshop of the smoke alarm production line, as shown in Fig 7. The model consists of a static attribute set, a process attribute set, and a functional component set.

   The static attribute set encompasses basic static information related to workshop operations, including workshop details (e.g., name, location, area, and responsible person), production organization (e.g., departments and workgroups), work calendar (e.g., working days and holidays), and production work orders (order information transmitted from the ERP system to MES).

   The process attribute set corresponds to dynamic workshop process data, typically comprising aggregated statistics from various workshop aspects and order tracking information used for feedback to the ERP system. This includes order tracking, production statistics, quality statistics, inventory statistics, and maintenance statistics.

                                                    

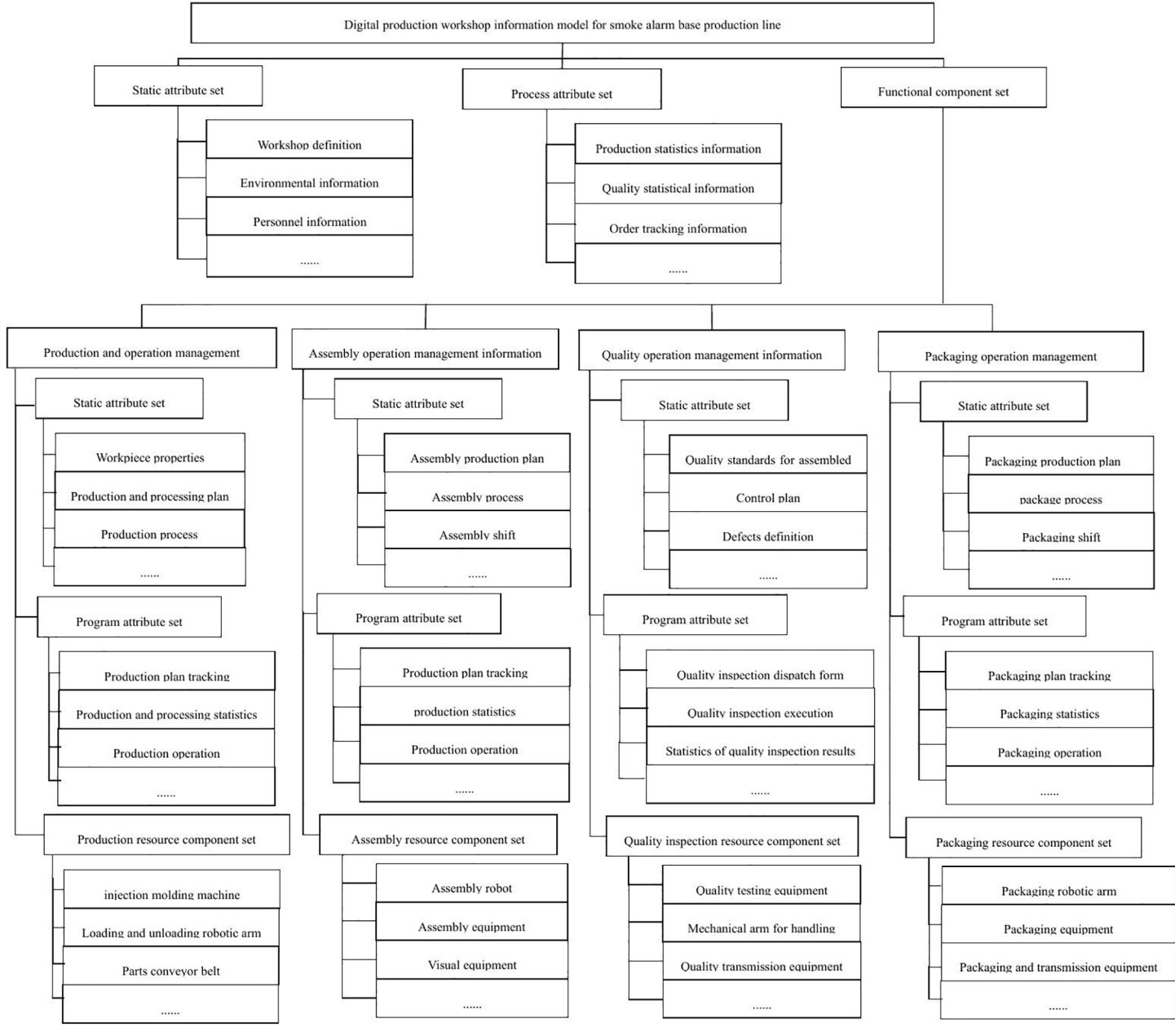

**Fig 7. Information model of digital production workshop for smoke alarm production line.**

The functional component set aligns with various production modules within the workshop, such as production, assembly, quality inspection, and packaging units. Each module contains a static attribute set, a process attribute set, and a resource component set. The static attribute set describes basic module information, including work plans, production techniques, and shift outputs. The process attribute set represents production process information for each resource module, including production plan tracking, operational data, and production statistics. The resource component set includes the various resources associated with each module, such as injection molding machines, industrial robots, and quality inspection equipment.

### 5.4. Data transmission realization

The integration of NX and OPC UA establishes a "virtual-real closed loop + abnormality robustness" data channel, meeting industrial-grade real-time requirements (data update latency ≤100ms), security, and reliability standards.

(1) OPC UA client configuration in NX

Enable the "OPC UA Client" module in NX MCD, connect to the OPC UA server of the edge gateway via the "Server Address", and adopt the "Basic256Sha256 encryption algorithm" to prevent data leakage and malicious tampering.

(2) Data subscription and publishing mechanism

The virtual model subscribes to physical device data (e.g., the "Injection Pressure" of a virtual injection molding machine is linked to the OPC UA server node ns = 1;s = InjectionUnit.InjectionMoldingMachine.InjectionPressure, with a subscription cycle of 50ms). Parameter adjustments during virtual commissioning (e.g., robotic arm speed) must first be evaluated by edge AI algorithms (such as collision detection and stress analysis) before being published to the physical PLC for execution, thereby mitigating commissioning risks.

(3) Exception handling and robustness enhancement

Heartbeat Detection: If no physical device data is received within 100ms, NX automatically marks the data as "invalid" and triggers an alarm (virtual device displays a red fault indicator).Multi-Level Caching and Data Repair: The edge gateway is equipped with 1GB of built-in cache to temporarily store data during network interruptions. Upon recovery, data is retransmitted based on timestamps. Simultaneously, AI models intelligently repair missing/abnormal data (e.g., interpolating missing values based on time-series patterns) to ensure data continuity.

## 6. Virtual-real mapping in the workshop

Virtual-real mapping serves as the cornerstone of digital twins. By establishing a multi-dimensional connection between virtual models and physical entities, it facilitates a closed loop of "physical world occurrence → data transmission → virtual world reflection → decision optimization → physical world execution." This study constructs a "multi-scale dynamic mapping error compensation theoretical model," dividing the mapping into four hierarchical levels and quantifying the technical objectives of each level (as shown in Table 2):

### 6.1. Implementation of geometric mapping

**6.1.1. Coordinate system unification.** Firstly, establish a global coordinate system for the workshop. The center of the injection molding machine nozzle is designated as the origin (0,0,0), with the X-axis running along the conveyor belt, the Y-axis perpendicular to it, and the Z-axis pointing upwards. Both physical equipment and virtual models are positioned

Table 2. Virtual real mapping hierarchy table of smoke alarm production line.

| Mapping hierarchy | Core content | Technical objectives |
|---|---|---|
| Geometric mapping | The virtual model corresponds to the shape, size and position of the physical entity | Geometric error ≤ 0.1 mm, position deviation ≤ 1 mm |
| Behavior mapping | The motion trajectory and action logic of the virtual model are consistent with the physical equipment | Action synchronization error ≤ 50ms |
| State mapping | The virtual model reflects the operation status of physical equipment in real time (such as operation/fault) | Status update delay ≤ 100ms |
| Parameter mapping | The process parameters (such as temperature and pressure) of the virtual model are synchronized with the physical equipment in real time | Parameter error ≤ 1% |

based on this coordinate system. A laser tracker (with an accuracy of ±0.05 mm) is utilized to calibrate the coordinates of the physical equipment, ensuring alignment with the virtual models. Secondly, consider dynamic geometric adjustments. For components prone to deformation, such as robotic arms, the "Finite Element Analysis" module in NX is employed to calculate stress-induced deformation. Real-time load data from the physical equipment is obtained via OPC UA, enabling dynamic adjustments to the geometry of the virtual model. For instance, when a robotic arm grasps a heavy object weighing 5 kg, the virtual model automatically simulates a 0.5 mm bending deformation based on the load signal from the physical sensor.

### 6.1.2. behavior mapping implementation.

(1) Kinematic mapping

The kinematic model of the manipulator is constructed using the D-H parameter method, and the mathematical relationship between the joint variables and the end effector pose is defined in the NX software. The joint angle data of the physical manipulator is obtained in real time through the OPC UA protocol and used to drive the rotation of the virtual manipulator's joints, ensuring that the virtual and actual motion trajectories are highly consistent, with a trajectory deviation controlled within 1 mm.

(2) Logical behavior mapping

The control logic of the virtual model is synchronized with the physical PLC program. The ladder logic program of the physical PLC is imported via the "PLC Open" interface in NX MCD to replicate logical control in the virtual environment (e.g., the logic of "injection completion → conveyor startup" in the physical PLC is implemented through a "signal trigger" mechanism in the virtual model). The edge AI model can anticipate motion conflicts (e.g., interference between a robotic arm and a conveyor) and preemptively adjust the control logic.

### 6.1.3. Implementation of state and parameter mapping.
A real-time synchronization strategy is employed. Firstly, timestamp-based synchronization is utilized, where physical devices attach precise timestamps (with a precision of 1 ms) when collecting data. Upon receiving the data, the virtual model aligns states and parameters according to these timestamps to avoid asynchronous discrepancies. Secondly, event-triggered synchronization is implemented; when a physical device experiences a state change (such as a fault or shutdown), an event signal (e.g., "Fault=1") is immediately sent out. The virtual model updates its state instantaneously upon receiving this signal (for instance, displaying a shutdown animation for the virtual device).

A deviation correction mechanism is adopted. Regular calibration is conducted every eight hours using laser interferometry to measure the actual position of the physical robotic arm and compare it with that of the virtual model. Deviations are calculated, generating correction parameters (e.g., X-axis compensation +0.02 mm), which are then written into the virtual model via OPC UA protocol. Additionally, dynamic compensation techniques are applied; specifically addressing the lagging characteristics of temperature in injection molding machine barrels, a temperature prediction model based on BP neural networks has been established within the virtual model. This model utilizes real-time temperature data from physical sources along with historical curves to proactively adjust virtual temperature values, ensuring that deviations between both remain within ±1°C.

## 6.2. Application scenarios based on mapping

### 6.2.1. Real-time monitoring and visualization.
In the NX virtual environment, a digital dashboard for the production line is constructed to display the operational status of each device in real time (green = running, yellow = standby, red = fault), along with output data (output per hour) and quality metrics (pass rate). Through virtual-physical mapping, the internal states of devices are visualized. internal conditions that cannot be directly observed in physical equipment (such as the screw position of an injection molding machine) are intuitively presented through the transparency feature of virtual models, assisting operators in assessing equipment status.

 

**6.2.2. Virtual debugging and process optimization.** Before launching a new production line, a full-process simulation is conducted within a virtual model.The reinforcement learning model automatically optimizes process parameters (e.g., robotic arm assembly speed, injection pressure-holding time combinations). Parameters for virtual debugging (such as robotic arm assembly speed) are transmitted to physical devices via OPC UA to validate process feasibility. To address quality fluctuations during smoke alarm assembly, adjustments are made to machine vision positioning parameters (e.g., template matching threshold) within the virtual model. The assembly deviations under different parameter settings are simulated, allowing optimal parameters to be applied back into the physical system through virtual-physical mapping—resulting in an increase in pass rate from 95% to 99.5%.

**6.2.3. Fault diagnosis and prediction.** When physical equipment experiences faults (e.g., robotic arm jamming), the virtual model reproduces fault phenomena through state mapping while simulating fault evolution processes using historical data (such as current changes prior to jamming) within a virtual environment—thereby identifying root causes of failures (e.g., bearing wear).

Based on parameter trend analysis derived from virtual-physical mapping, A long-term record of clamping force variation curves for injection molding machines is maintained by the virtual model; when these curves exhibit a slope exceeding predetermined thresholds (for instance, a monthly decline greater than 2%), it predicts mold wear and generates maintenance alerts proactively.

## 7. Case study and system validation

### 7.1. Application scenarios

Currently, the production of smoke alarms at an electronic technology company primarily relies on manual labor. The outdated method of integrating workshop production data, coupled with a lack of effective monitoring tools, results in low transparency and difficulty in managing workshop operations, significantly hindering production efficiency. To address these issues, the company is advancing the intelligent upgrading and transformation of its workshops, with the goal of enhancing the intelligence of production lines, optimizing workshop management, and increasing workshop transparency. The digital twin workshop architecture and four key technologies presented in this paper offer an effective solution for the smoke alarm production line.

By utilizing the twin model modeling approach, data acquisition and transmission solutions outlined in this paper, along with the process flow of the actual production line,we have developed a digital twin model for the smoke alarm production line and established a virtual workshop. Powered by real-time data, the virtual workshop precisely mirrors the production and processing activities in the physical workshop.Building on this foundation,we have deployed and implemented a three-dimensional, visually enhanced real-time monitoring system in the workshop, leveraging digital twin technology, to enable real-time tracking of production status and assist staff in comprehensively understanding the workshop's production status and operational information. Fig 8 displays the main interface of the twin system.

There are four main buttons at the top of the main interface, i.e., start, reset, pause and stop, which can manually control the start, pause, stop, reset and other functions of the production line. There is a current alarm display column under the four buttons. If the production line fails, it will scroll through the interface. The real-time output is displayed on the left side of the middle position of the main interface in real time. The simulation operation module can simulate key unit operations such as injection molding, assembly, packaging, and reclaiming before actual production. The right reset initialization status module manually initializes the main nodes of the production line process. At the bottom of the main interface, there are five button modules: manual page, alarm page, parameter page, debugging page and main page. The manual page can realize the function of manually operating the production line, the parameter page can display the production parameters of each equipment of the production line in real time, and the debugging page can debug each unit of the production line.

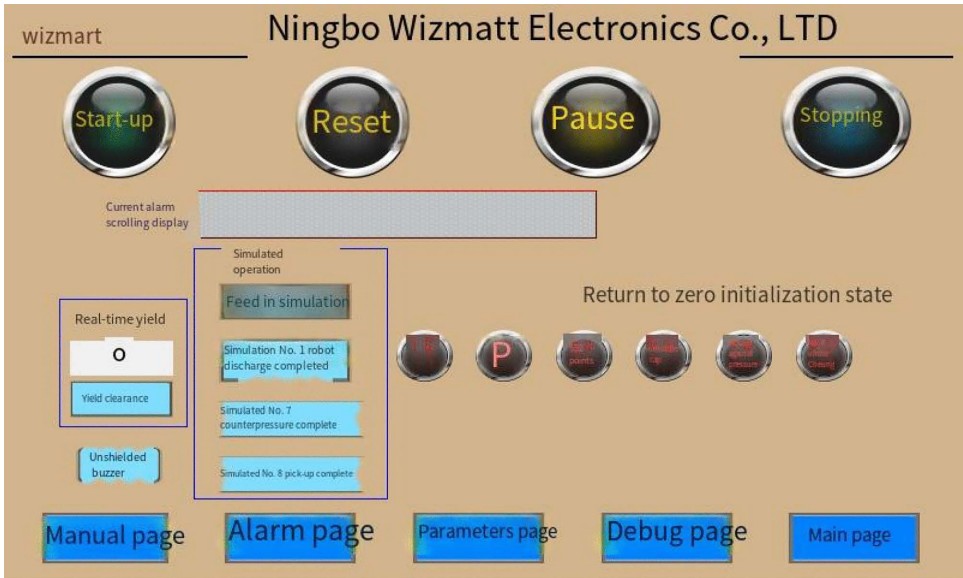

**Fig 8. Twin system main interface.**

Fig 9 shows the workshop monitoring interface. The middle position of the interface displays the working data of the production line in real time, and the corresponding equipment position displays the working status of the equipment in real time. Green indicates normal operation, yellow indicates standby, and red indicates fault alarm. The upper left corner of the interface displays the humidity, temperature, startup rate, gas supply pressure, utilization rate and other data of the workshop in real time. The upper right corner can display the alarm information of the production line in real time. An emergency stop button is set at the lower part of the interface, which can be manually stopped in case of emergency. The machine vision detection information that cannot be seen in the model is displayed at the bottom of the interface in real time. At the bottom right of the interface is the workshop data summary chart, which can display the workshop running time, product qualification rate summary, workshop output and other data in real time.

After meticulous on-site debugging and practical implementation,the system has demonstrated exceptional performance.It operates extremely smoothly, maintaining a stable average response time of 48.6ms (±3.2ms), fulfilling the real-time demands for system data acquisition and transmission. Currently, a synchronous and efficient operational state has been largely achieved between the twin (virtual) workshop and the physical workshop.

The digital twin system primarily fulfills three key functions. In terms of tracking production progress, the system is capable of precisely depicting the real-time advancements of each stage in the smoke alarm production line, spanning from raw material input to finished product output. The completion status and time nodes of each step are clearly discernible, enabling staff to promptly identify production bottlenecks, agilely adjust resource allocation, and guarantee a seamless and unimpeded production flow.

In terms of equipment status monitoring, the system provides comprehensive surveillance of all types of equipment on the production line, real-time collecting operational parameters such as temperature, rotational speed, and vibration. Upon detecting any abnormal conditions in the equipment, the system promptly issues early warnings to alert maintenance personnel to potential failure risks in advance, enabling them to arrange for preventive maintenance. This effectively minimizes the equipment failure rate and mitigates production disruptions caused by equipment shutdowns.

For quality control, by conducting an in-depth analysis of production process data and correlating it with product quality data, we can precisely pinpoint the stages and causes of quality issues. For instance, if welding defects are found in a

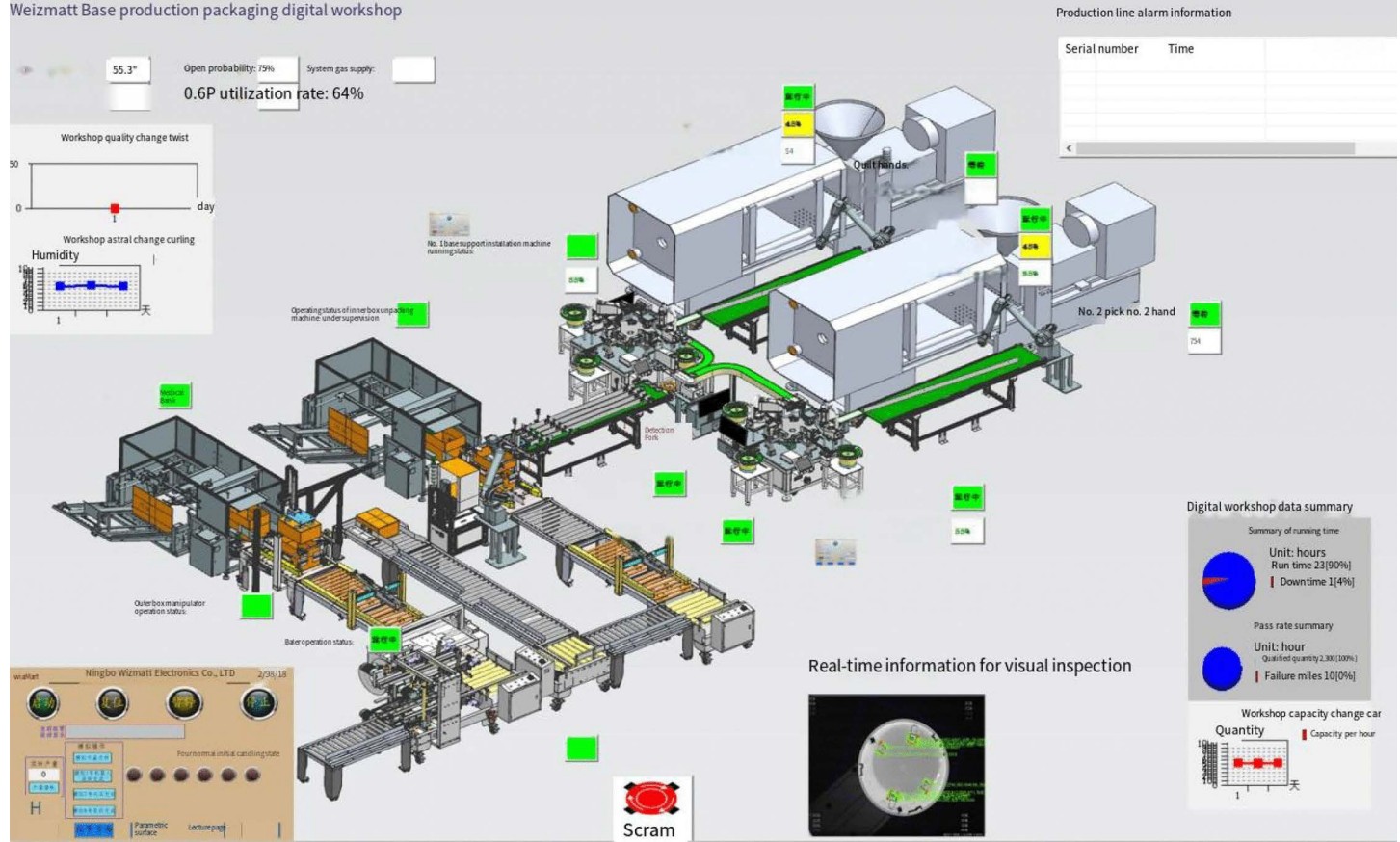

**Fig 9. System monitoring interface.**

batch of smoke alarm, the system can swiftly trace these back to welding equipment parameters, operators, and environmental factors at the time, furnishing a solid foundation for quality enhancement and aiding enterprises in elevating their product quality.

In addition, the system boasts energy management capabilities, enabling real-time monitoring of energy consumption across the workshop and analysis of energy efficiency for each piece of equipment and production stage. Leveraging intelligent algorithms, it offers energy optimization recommendations for workshops, assisting enterprises in reducing energy consumption and achieving green production. Thanks to these functions, the digital twin monitoring system significantly enhances the intelligence level and management efficiency of the smoke alarm production line, thereby generating greater value for enterprises.

## 7.2. Operational performance comparison

The core theoretical contribution of this paper lies in proposing and validating a "Full-Process Digital Twin with Tightly Coupled Physical Systems" architecture. Through the organic integration and closed-loop feedback between its automation execution and digital twin intelligence layers, this architecture overcomes the limitations of existing technologies.Its fundamental difference from current production technologies achieves a paradigm shift from "single-point automation optimization" to "end-to-end data-driven intelligence," rather than merely replacing equipment or simply layering technologies.

The traditional production process is as follows: The injection molding machine first forms the base component of the smoke alarm. This is then manually sorted before being transported to the assembly stage. On the assembly line, workers assemble several metal parts with the base, followed by another round of manual sorting. Subsequent steps include manually folding cardboard boxes, packing assembled products into boxes, folding outer cartons, manually applying plastic edge sealing, bundling, and palletizing. The entire process requires 30 operators to meet production capacity demands.

The existing production line has undergone automation upgrades, significantly optimizing processes: After injection molding, base components are automatically unloaded by robotic arms and conveyed via conveyor belts to assembly equipment. Robots then pick up products for visual inspection. Following assembly of metal parts, a secondary visual inspection is conducted. Qualified products are then conveyed to automated packing equipment. A case opener automatically unfolds and positions the cardboard box, into which a robot automatically places the product. Subsequently, products sequentially travel via conveyor belt to a case sealer for primary and secondary sealing, then enter an edge sealer for automatic edge sealing. They are then automatically bundled by a strapping machine and finally palletized. The entire production line requires only one operator to manage auxiliary material replenishment for continuous operation.

It is precisely on this foundation of automation, combined with the deeper integration of digital twin technology, that we have achieved a leap from "equipment replacing human labor" to "data-driven intelligence." This has resulted in significant differences from traditional production models and existing partial automation technologies across multiple dimensions.

(1)  Core theoretical innovations

Herein, we propose the "Full-Process Digital Twin with Tightly Coupled Physical Systems" architecture. This work constructs a five-tier, closed-loop intelligent production framework encompassing: the Physical Production Layer, the Data Acquisition and Transmission Layer, the Virtual Simulation and Modeling Layer,the Intelligent Decision and Optimization Layer, and the Execution and Feedback Layer. This architecture faithfully mirrors all production elements (equipment, process, material, quality, personnel) and facilitates their real-time dynamic interaction. It directly addresses the critical limitations of traditional systems, the decoupling of physical entities from their virtual models and obstructed data flow,thus providing a solid theoretical basis for holistic process optimization and intelligent production control.

We established a data-driven predictive control model founded on full-link, multi-source heterogeneous time-series data, including injection molding temperature, pressure, assembly force, equipment vibration, and visual inspection features. A suite of multi-parameter coupled models for quality prediction and equipment health assessment is proposed and validated. This model proactively prevents quality issues, shifting control from a passive "detect-and-react" mode to an active "predict-and-prevent" mode. It also replaces "reactive" or "preventive" maintenance with data-driven predictive maintenance, thereby significantly boosting system stability and reliability.

We established a data-driven predictive control model founded on full-stream, multi-source heterogeneous time-series data, including injection molding temperature, pressure, assembly force, equipment vibration, and visual inspection features. A suite of multi-parameter coupled models for quality prediction and equipment health assessment is proposed and validated. This model proactively prevents quality issues, shifting control from a passive "detect-and-react" mode to an active "predict-and-prevent" mode. It also replaces "reactive" or "preventive" maintenance with data-driven predictive maintenance, thereby significantly boosting system stability and reliability.

We designed an adaptive scheduling mechanism tailored for flexible production. Through the virtual simulation layer of the digital twin system, an algorithm based on genetic algorithms and reinforcement learning is introduced to iteratively optimize process parameters and production schedules. This capability allows for rapid simulation of production flows across various product specifications, optimization of parameter sets, and automatic generation of the most efficient production plan. Consequently, it drastically cuts down on debugging time and costs in real-world changeover scenarios, directly tackling the inherent pain points of traditional automated lines: "inflexibility, high changeover costs, and poor market responsiveness."

(2)   Fundamental differences from existing technologies

Compared to existing related technologies, the method proposed in this paper achieves fundamental breakthroughs in the following aspects:

Distinction from mere automation transformation. Existing mere automation transformations (such as simply introducing robotic arms or automated conveyor lines), while capable of replacing manual operations to a certain extent and improving production efficiency, essentially represent a "rigid" production mode. The production process lacks effective real-time monitoring and dynamic optimization capabilities, and the correlations between equipment status, process parameters, and product quality are difficult to deeply explore and utilize. The overall production system remains in a "black box" state, with decisions primarily based on experience. In contrast, our method, centered on the "Full-Process Digital Twin with Tightly Coupled Physical Systems," achieves complete "transparency" and "predictability" of the production process through digital twin technology. This elevates the system to the level of "data-driven intelligent decision-making," realizing a qualitative leap from "replacing human labor" to "globally optimizing the entire production chain."

Distinction from isolated digital twin applications.Currently, the application of digital twin technology in manufacturing is mostly confined to individual processes or single pieces of equipment,for instance, being used only for simulation and optimization of the injection molding process, or for health monitoring of a specific critical machine. Such isolated application modes result in "information silos" of data, where information from different stages cannot be effectively integrated or shared, making it difficult to form a holistic understanding or achieve system-wide optimization of the entire production process. In contrast, the method proposed in this paper extends digital twin technology across the entire workflow, from product molding and component assembly to quality inspection, packaging, and palletizing, encompassing the full sequence of "molding – assembly – inspection – packaging – palletizing." This enables seamless data flow and deep integration across all stages. The virtual model can receive real-time feedback from the physical layer and dynamically adjust and optimize accordingly, thereby guiding the operation of the physical system in reverse and forming a complete closed-loop optimization cycle.

Distinction from traditional data-driven production.Traditional data-driven production systems primarily focus on collecting, storing, and performing basic statistical analysis of production data. Their main purpose is often limited to post-event traceability and report generation. A key limitation is their general lack of capability for virtual simulation, scenario modeling, and predictive analysis using the collected data.Consequently, the decision-making process in such systems remains heavily dependent on the experiential judgment of managerial personnel. This reliance on human intuition introduces subjectivity and limits proactive optimization.In clear contrast, our method fully utilizes the virtual simulation and modeling power of digital twin technology. We construct a virtual production system that maintains high fidelity with the physical world. This virtual environment allows for the simulation, testing, and optimization of various production scenarios without disrupting actual operations. The optimal strategies identified in this virtual space are then deployed directly to the physical system.This creates a continuous "trial-and-error – optimization" cycle grounded in virtual simulation. When integrated with machine learning algorithms, this cycle enables the self-optimization of process parameters, self-traceability of quality issues, and self-prediction of equipment failures. Ultimately, this approach achieves truly intelligent, autonomous operation and facilitates the continuous improvement of the production system.

(3)   Quantitative validation and technical advantages (comparison between traditional production models and the method presented in this paper)

Based on the aforementioned theoretical innovations, the modern digital twin production line (technology-driven + data intelligence) has fundamentally transformed traditional production models.

Core limitations of traditional production models (labor-intensive):

Production efficiency, The entire manufacturing process relies heavily on manual labor, with over 15 major operations performed by hand. Production cadence is severely constrained by factors such as worker skill levels, labor intensity, and

physical fatigue. This results in a lengthy overall production cycle per unit, creating significant bottlenecks for capacity expansion and limiting daily output to a low ceiling.

Labor costs, to meet basic production capacity requirements, the production line must be staffed with at least 30 operators. Calculated at a comprehensive monthly labor cost per person of approximately 6,000 yuan (including wages, social insurance, management, etc.), direct labor costs alone amount to 180,000 yuan per month. This represents not only a substantial ongoing expense but also comes with numerous challenges in recruitment, training, and management. As labor costs rise year after year, this burden will become increasingly heavy.

Quality costs, quality inspection relies entirely on manual visual screening, resulting in relatively high rates of missed defects and misidentifications. This keeps the product defect rate at a persistently high level (2%–5%). Not only does this directly lead to elevated rework costs and potential material waste, but it may also trigger customer complaints and damage the company's reputation. More critically, when quality issues arise, the lack of effective data recording and traceability makes it difficult to quickly pinpoint the root cause.

Management models, the production process resembles a "black box," making it difficult for managers to gain real-time, accurate insights into actual conditions on the production floor. Management decisions primarily rely on experience, lacking scientific and precise data support, which results in inefficient production scheduling and resource allocation.

The practical implementation results of the methods described in this paper:

Production efficiency, automated equipment ensures 24/7 continuous operation at a fixed, high-speed cycle. The digital twin system further amplifies this gain by using real-time data and virtual simulation to dynamically optimize the entire process, achieving an efficiency boost of 400% to 600% compared to the conventional model. Process debugging and planning in the virtual environment have also reduced changeover time by over 50%, which in turn greatly increases production flexibility and enables a faster response to diverse market needs.

Labor costs, the entire complex production process requires only one employee to handle auxiliary material addition and monitor system operation status. Direct labor costs plummeted to approximately 6,000 yuan per month, achieving monthly savings of up to 174,000 yuan compared to traditional methods—a 96.7% reduction in labor expenses. The employee's role has also evolved from a traditional manual laborer to a modern technical administrator and system maintainer.

Quality control, the introduction of robotic vision systems enables two rounds of 100% comprehensive precision inspections, ensuring consistent product quality. Concurrently, digital twin technology monitors and records critical process parameters throughout production(such as injection molding temperatures and press-fitting forces)in real time, achieving full traceability of product quality. Should any quality issues arise, they can be swiftly traced to specific production stages, equipment parameters, or even raw material batches, enabling predictive quality control. Consequently, the product defect rate has been significantly reduced to below 0.1%.

Value-added benefits of the digital twin.Beyond the direct benefits outlined above, digital twin technology delivers substantial additional value.By virtually mapping physical production lines in real time, it enables real-time equipment monitoring and predictive failure detection. Predictive maintenance strategies based on data analysis have effectively reduced unplanned downtime by over 70%. By simulating and testing different combinations of process parameters, continuous optimization of production processes has increased overall equipment effectiveness (OEE) by 5% to 15%. The system supports remote monitoring and decision-making, significantly enhancing management efficiency and quality control capabilities, particularly for group-wide and cross-regional production management.

Comprehensive cost and return on investment.Although the initial investment for establishing a digital twin production line is relatively high-including sensors, IoT devices, data acquisition and processing systems, digital twin platform software, and related integration services-this investment is typically recovered within 2–4 years. This is achieved through drastic reductions in labor costs, quality-related costs, and equipment downtime losses. Beyond the payback period, the system continues to generate significant profit gains. Moreover, due to its high flexibility and scalability, the company is endowed with stronger capabilities for scalable expansion and enhanced market competitiveness.

The theoretical innovations and practical implementations in this study demonstrate a core value that goes beyond mere "human-to-machine substitution" through automation. More significantly, by constructing a "Full-Process Digital Twin with Tightly Coupled Physical Systems" architecture, we have achieved an integrated operational model in smoke alarm production for the first time. This model seamlessly combines "Digital Twin Mirroring," "Data Closed-Loop Driving," and "Predictive Intelligent Control" across the entire workflow—from product design and process planning to production execution and after-sales service.

This breakthrough addresses the common limitations of existing technologies, such as isolated point optimization, the virtual-physical disconnect, and rigid production structures. Consequently, it provides a replicable theoretical framework and a validated technical pathway for discrete manufacturing. It specifically facilitates the transition from traditional labor-intensive models to a modern "technology- and data-dual-driven" intelligent production paradigm, supported by quantitative evidence of its success.

## 8. Conclusion and discussion

As a critical bridge connecting the physical world and the virtual space, digital twin technology provides core support for the digital transformation of the manufacturing industry. To address the digitalization needs of a smoke alarm production line, this study constructed a hierarchical digital twin system encompassing "device-level, unit-level, and production line-level" layers. A multi-source heterogeneous data acquisition and transmission architecture based on OPC UA was designed and successfully implemented in a production line of an enterprise in Ningbo. Practical application demonstrated that the system operates stably and achieves core functions such as real-time production progress tracking, full-process quality control, and online equipment status monitoring. Compared with the traditional production mode, the production efficiency increased by 400%–600%, the product defect rate was significantly reduced to below 0.1%, and labor costs decreased by 96.7%. The following section will discuss the rationality of the results, improvement mechanisms, and research limitations.

(1)  Analysis of result rationality and improvement mechanisms

The improvement in the defect recognition rate stems from the fundamental enhancement of traditional detection methods by the "virtual-real mapping" mechanism of the digital twin. Traditional production lines rely on manual sampling inspections and machine vision based on fixed thresholds, which are prone to issues such as missed detections (e.g., fine scratches) and misjudgments (e.g., grayscale deviations caused by lighting interference). This study achieved higher precision and robustness in quality control by establishing a virtual-real interactive defect recognition mechanism:

The virtual model utilized the NX Open API to call the OpenCV algorithm library, simulating the grayscale characteristics of various defects (e.g., pores and dimensional deviations) in the virtual environment, thereby forming a dynamically updated defect template library.

Images captured by the physical vision system were preprocessed by edge nodes and then compared and bidirectionally validated with the virtual template library in real time, effectively suppressing misjudgments caused by environmental interference.

This mechanism increased the defect recognition rate from 96.3% to 99.5% (a net increase of 3.2%) while keeping the misjudgment rate below 0.1%, demonstrating the precision and reliability of the digital twin in quality control.

The reduction in energy consumption was primarily attributed to the digital twin's capability for dynamic optimization of equipment status and process parameters:

Equipment data (e.g., injection molding machine barrel temperature and robotic arm motor speed) were collected in real time via OPC UA, and an LSTM prediction model was employed to identify abnormal energy consumption. For instance, when the barrel temperature exceeded the process requirement by 2°C, the system automatically issued commands to adjust the heating power, saving approximately 1.2 kWh per hour for a single device.

The production line-level virtual model optimized logistics scheduling through path simulation and virtual commissioning, such as reducing the AGV transportation path by 15% to minimize no-load and ineffective energy consumption. Additionally, equipment operational status was dynamically adjusted based on order loads, avoiding energy waste caused by "overpowered equipment handling minor tasks."

The above mechanisms collectively reduced the overall energy consumption of the production line by 8% and maintained stable energy efficiency even under order fluctuations (daily production volume variations of ±20%).

The improvement in production efficiency primarily relied on virtual commissioning replacing physical trial-and-error and dynamic scheduling through virtual-real collaboration:

Traditional production lines required 停机调试 (e.g., adjusting robotic arm trajectories) during process switches, taking an average of 4–6 hours. This study used the virtual environment to simulate debugging processes in advance, verifying parameter feasibility and reducing actual debugging time to within 30 minutes, shortening the single-batch cycle by 20%.

Bottleneck visualization is achieved through virtual-real mapping: the virtual interface displays the status of each workstation in real time (e.g., triggering an alert when assembly waiting time exceeds 5 minutes). The scheduling system dynamically optimizes material distribution based on this information, increasing the production line balance rate from 78% to 92%, and ultimately improving overall efficiency by 15.3%.

## (2)   Research limitations

Although the system demonstrated significant results in practical applications, the following limitations remain.

Deviation in model accuracy under dynamic working conditions: The virtual model exhibited errors in geometric and behavioral mapping under high-load scenarios. For example, when the robotic arm grasped objects weighing over 5 kg, the actual arm bending deformation (approximately 0.8 mm) deviated from the finite element prediction (0.5 mm) by 0.3 mm. The primary cause was the virtual model's insufficient accounting for the reduction in elastic modulus due to accumulated material fatigue.

Bottleneck in edge node processing capability: When the current edge gateway (Siemens IoT 2050) processed high-frequency concurrent data from multiple devices (e.g., device status at 100 ms intervals and process parameters at 50 ms intervals), the data compression rate dropped from 90% to 75%. This resulted in transmission delays exceeding 100 ms for some non-critical data (e.g., environmental temperature and humidity), affecting the real-time synchronization efficiency of the virtual model.

The integration of blockchain introduces a cost-efficiency trade-off: while the consortium blockchain consensus mechanism (e.g., practical byzantine fault tolerance) ensures the immutability of critical data (e.g., quality inspection results, reducing the tampering rate to zero), it also introduces an approximate 20 ms data write latency and increases node maintenance costs by about 15% compared to conventional architectures. In small and medium-batch production scenarios, its economic viability remains to be optimized.

## (3)   Summary and outlook

This study validated the feasibility of digital twin technology in the digital transformation of smoke alarm production lines. Its core value lies in breaking the "black box" of traditional manufacturing through a "virtual-real closed loop," achieving a fundamental shift from experience-driven to data-driven operations. Future work can be pursued in the following three aspects:

Introduce a material fatigue damage model to enhance prediction accuracy under high-load conditions.

Adopt an edge node cluster architecture to improve the processing capability for high-frequency concurrent data.

This study provides a reusable architectural and methodological framework for constructing digital twin systems in discrete manufacturing, though its generalizability and scalability require further validation in broader manufacturing scenarios, such as electronic assembly and automotive components.

## Author contributions

**Data curation:** Wenfeng Ying.

**Writing – review & editing:** Min Wu.

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
