## [Decision Letter · Decision Letter 0]

5 Aug 2025

Dear Dr. Wu,

Thank you for submitting your manuscript to PLOS ONE. After careful consideration, we feel that it has merit but does not fully meet PLOS ONE’s publication criteria as it currently stands. Therefore, we invite you to submit a revised version of the manuscript that addresses the points raised during the review process.

We look forward to receiving your revised manuscript.

Kind regards,

Himadri Majumder, Ph.D

Academic Editor

PLOS ONE

4. We notice that your supplementary figures are included in the manuscript file. Please remove them and upload them with the file type 'Supporting Information'. Please ensure that each Supporting Information file has a legend listed in the manuscript after the references list.

Reviewers' comments:

Reviewer's Responses to Questions

**Comments to the Author**

1. Is the manuscript technically sound, and do the data support the conclusions?

Reviewer #1: Yes

2. Has the statistical analysis been performed appropriately and rigorously?

Reviewer #1: I Don't Know

3. Have the authors made all data underlying the findings in their manuscript fully available?

Reviewer #1: No

4. Is the manuscript presented in an intelligible fashion and written in standard English?

Reviewer #1: Yes

Reviewer #1: The article proposes a method for constructing a digital twin workshop for the production line of smoke alarm bases, designed through a four-dimensional architecture comprising the physical layer, model layer, data layer, and service layer. It integrates multi-domain modeling techniques (geometry, physics, behavior, and information) and achieves multi-source heterogeneous data integration and virtual-real collaborative mapping based on the OPC UA protocol. A 3D visualization monitoring system was implemented to enable traceability verification of the production process. Experimental results indicate that the system can reduce labor costs by over 2.1 million yuan and increase product output by 2%. The article is well-structured, but its innovation is limited, and several issues remain:

1. The literature review focuses on industrial applications of digital twins but does not mention case studies comparing related technologies in other low-automation industries (e.g., traditional manufacturing). It is recommended to include discussions of studies in similar contexts.

2. The description of digital twin model construction in Section 3 relies heavily on mathematical expressions for definitions. While these equations provide a theoretical framework, they are highly abstract, lacking derivations of variables and their applications, as well as specific details on how they are quantified or implemented in Siemens NX software. The authors should provide concrete examples and relevant descriptions.

3. In Section 4, the case study validation mentions cost savings (2.1 million yuan/year) and a 2% output increase but lacks control group data (e.g., comparisons with traditional production lines) and corresponding details. It is suggested to include details of the experimental design and statistical analysis methods.

4. Figures 7 and 8 are critical for demonstrating the system's actual implementation, but the text lacks descriptions of specific metrics in the interface (e.g., production progress, equipment status). For example, providing sample outputs or screenshots to illustrate the interface's functionality would be helpful. The authors should provide detailed information explaining the key elements depicted in each figure.

5. The figures provided in the article are mostly conceptual, focusing on architecture and models, with a lack of actual data and real-world scenarios.

6. Transitions between sections are weak. For instance, the transition from model construction (Section 3.1) to information system design (Section 3.2) lacks connective narrative. The authors should add brief transitional paragraphs to clarify how each section builds on the previous one, thereby improving the overall coherence of the manuscript.

**Do you want your identity to be public for this peer review?** For information about this choice, including consent withdrawal, please see our Privacy Policy

Reviewer #1: No

---

## [Author Response · Author response to Decision Letter 1]

17 Aug 2025

Dear Editor and Reviewers,

We thank you very much for giving us another chance to revise and improve our manuscript. We try our best to revise the paper according to the reviewers’ suggestions, and for those comments we have different opinions, we have explained in detail. A list of changes and responses are listed below.

We hope the revised paper has addressed all the reviewers’ concerns and reached the requirements of the journal for publishing.

Thank you very much!

The authors

Responses to Reviewers’ Comments

Reply to reviewer #1

1.Concern of the reviewer:

The literature review focuses on industrial applications of digital twins but does not mention case studies comparing related technologies in other low-automation industries (e.g., traditional manufacturing). It is recommended to include discussions of studies in similar contexts.

Our response: We are grateful to the Reviewer for his elaborate work to find out our weakness.We have added the comparison between digital twin technology and traditional manufacturing .

Revised text:

At the same time, many scholars have made a comparative study of digital twin technology and traditional manufacturing technology. Semeraro, et al. [9] proposed a data-driven digital twin modeling method to improve the robustness of the model through the invariance mode, and compared with the flexibility and scalability of traditional rule-based modeling methods in complex production systems. Pronost, et al. [10] systematically reviewed the application of digital twins in the whole product life cycle (design, production, operation and maintenance), pointed out the limitations of traditional simulation technology in real-time and cross scale integration, and emphasized the advantages of digital twins in dynamic optimization. The case showed that the product development cycle was shortened by 28%. Mayer, et al. [11] analyzed the technical trend of digital twins in manufacturing, compared the limitations of traditional simulation tools in multi physical field coupling modeling, proposed the advantages of digital twins in interdisciplinary collaboration, and quantified its improvement in the modeling accuracy of complex systems by 42%. Fu Xiangfu, et al.[12] aimed at the limitations of traditional processing error research methods, proposed a digital twin driven multi-scale quality representation framework, and compared with the traditional off-line detection method, verified that its accuracy in real-time error compensation was improved by 51%. D'ambrogio, et al.[13] reviewed the application of AI driven digital twins in manufacturing, compared with the traditional threshold based maintenance strategy, and proposed the dynamic decision-making advantage of digital twins in predictive maintenance. The case showed that the downtime of equipment was reduced by 34%. Pathy, et al. [14] built a supply chain optimization model driven by digital twins, and compared with the traditional static planning method to verify its robustness under dynamic disturbance. The experiment showed that the logistics cost was reduced by 29%. Park, et al. [15] developed a digital twin driven optimization system for production line design. Compared with the limitations of traditional simulation software in layout design, the case shows that the reconstruction time of production line is reduced by 40%. Through specific cases and data comparison, the above papers have revealed the significant advantages of digital twin in design cycle, cost control, real-time optimization, and pointed out the shortcomings of traditional manufacturing technology in complex system management and dynamic response.

2.Concern of the reviewer:

The description of digital twin model construction in Section 3 relies heavily on mathematical expressions for definitions. While these equations provide a theoretical framework, they are highly abstract, lacking derivations of variables and their applications, as well as specific details on how they are quantified or implemented in Siemens NX software. The authors should provide concrete examples and relevant descriptions.

Our response: Thanks very much to the Reviewer for his meticulous work to find out our weakness , To solve the above problems, in order to make readers more clearly understand the production content of the smoke alarm base, we added the section on the physical system composition and process flow of the base production line, adjusted the original section 3 to section 4, and added the section 4.4 on the construction of NX twin model in section 4, which described in detail how to build the twin model in NX.

Revised text:

3. Physical system composition and process flow of the base production line

3.1 Physical system composition

The production line for smoke alarm bases comprises five core subsystems (as depicted in Figure 2). The equipment and functions of each subsystem are detailed below:

Injection Molding Subsystem: this includes one horizontal injection molding machine (with a clamping force of 500 KN), a raw material dryer, and a cooling water circulation system. It is responsible for melting, injecting, pressure-holding, and cooling the plastic raw materials to form the base.

Assembly Subsystem: comprising two six-axis robots (with a repeat positioning accuracy of ±0.02 mm), a conveyor belt (operating at a speed of 0.5 m/s), and tooling fixtures, this subsystem handles the assembly of the base and metal inserts.

Machine Vision Subsystem: this subsystem includes two industrial cameras (each with 5 million pixels), lenses (with a focal length of 16 mm), an annular LED light source, and an image processing unit. It is used for assembly positioning guidance and quality inspection.

Detection Subsystem: equipped with size detection sensors (with an accuracy of ±0.01 mm) and appearance defect detection devices, this subsystem conducts a comprehensive inspection of the assembled bases.

Packing and Sealing Subsystem: this subsystem includes an automatic packing machine, a tape sealing machine, and a labeling machine, which together facilitate the packaging and identification of qualified bases.

Figure2 Physical drawing of workshop

3.2 Process flow analysis

The entire production process of the base can be categorized into four crucial stages, with the following process parameters and logical relationships for each:

Injection Molding Stage: raw materials undergo drying (at a temperature of 80°C for 4 hours) before being fed into the injection molding machine's barrel. Upon heating (with a nozzle temperature of 220°C), they melt and are injected into the mold cavity. The pressure is maintained at 80 MPa for 3 seconds, followed by a cooling period of 15 seconds. Once the mold is opened, the base blank is ejected.

Manipulator Assembly Stage: the conveyor belt transports the base blank to the assembly station, where the machine vision system captures an image of the base's positioning holes. After calculating any deviations, the manipulator is guided to grasp the metal insert (with a diameter of 8 mm) and press it into the preset position on the base (with a pressing force of 500 N).

Visual Inspection Stage: an industrial camera at the inspection station captures 360° images of the assembled base. Through image algorithms (combining template matching and edge detection), it assesses the perpendicularity of the insert assembly (with a tolerance of ±0.1 mm) and detects any appearance defects, such as cracks or material shortages.

Packing and Sealing Stage: qualified bases are conveyed to the packing machine via a conveyor belt and stacked in boxes of 20 each. After the sealing machine applies tape to the top and bottom of the box, a labeling machine prints and affixes a QR code label containing the production date and batch information. Each stage is logically controlled by a PLC (Siemens S7-1200), with state feedback provided by sensors (proximity switches and pressure sensors), thereby forming a continuous production process.

4.4 Construction based on NX twin model

The construction of the twin model for the smoke alarm base production line adheres to a hierarchical approach of "equipment level → unit level → production line level," with the specific framework outlined below:

Equipment Level Model: establish geometric and behavioral models for individual equipment such as injection molding machines, manipulators, and cameras.

Unit Level Model: integrate equipment-level models into subsystems (e.g., injection molding, assembly) to facilitate collaborative simulation among equipment within these subsystems.

Production Line Level Model: combine unit-level models to create a virtual representation of the entire production line, enabling comprehensive process simulation and optimization.

he modeling process must adhere to the three principles of "geometric consistency, behavioral equivalence, and parameter correlation" to guarantee that the virtual model accurately mirrors the shape, motion, and performance of the physical entity.

4.4.1 Equipment level model construction

(1) Modeling of injection molding machine

Geometric modeling:

Using NX's "synchronous modeling" technology, the 3D skeleton model is generated by importing the 2D drawing (CAD format) of the injection molding machine.

Decompose the injection molding machine into sub components such as clamping mechanism, injection mechanism and ejection mechanism, and use "Boolean operation" and "feature modeling" (such as stretching and rotation) to build the detail structure.

The accuracy of key components (such as screw and mold cavity) is controlled at 0.005mm, and non key components (such as shell) can be simplified (removing chamfer, small hole and other features) to reduce the complexity of the model.

Behavior modeling:

Define the motion pair in NX MCD: add "rotation pair" (opening and closing mold rotation axis) and "translation pair" (template movement) to the clamping mechanism, and add "screw pair" (screw rotation and axial movement) to the injection mechanism.

Associated driving parameters: drive the motion pair through the "servo motor", set the mold opening and closing speed (0.5m/s), injection pressure (0-100Mpa) and other parameters, corresponding to the PLC control parameters of the physical equipment.

Add sensor model: add "position sensor" (to detect the opening and closing state of the mold) and "pressure sensor" (to simulate the cavity pressure) at the mold cavity position, and its signal output is bound to the state variable of the virtual model.

(2) Manipulator and machine vision model construction

Geometric modeling of manipulator:

The "top-down" design method is adopted. Firstly, the base and joint skeleton of the manipulator are constructed, and then the arm, wrist and end effector are added through "assembly constraints" (such as coincidence and parallel).

The end effector (gripper) adopts parametric design, and the opening and closing size of the gripper is associated through "expression" (to adapt to inserts with a diameter of 5-10mm).

Robot behavior modeling:

Add a "rotation pair" at the joint to set the rotation range (-180 °~180 °) and angular velocity (50 °/s).

Through the "path planning" function of NX, predefine the motion tracks of grabbing, moving and placing (adopt arc transition to avoid emergency stop).

Associate machine vision signals: install a virtual camera model at the wrist, set the field of view angle (60 °) and resolution (5million pixels), and the output "positioning deviation value" is used as the compensation parameter of the manipulator movement.

Machine vision system modeling:

Build the geometric model of camera, lens and light source in NX, and define the installation position through the "coordinate system" (consistent with the installation coordinates of the physical camera).

Develop virtual image processing module: call opencv algorithm library through NX open API, realize "template matching" (locating the hole position of the base) and "defect detection" (simulating gray value analysis) in the virtual environment, and output the detection results (ok/ng signal).

(3) Construction of packing and sealing equipment model

Geometric modeling:

The sealing machine is decomposed into conveying roller table, pressing roller and belt cutting mechanism, and the "sheet metal design" module is used to build a thin shell structure (1mm thick).

The carton model adopts "folding design" to simulate the deformation characteristics of the carton through the "flexible body" attribute.

Behavior modeling:

Add "rotating pair" to the conveying roller table and set the speed (100r/min).

The "translation pair" is added to the pressing roller mechanism, and the pressing stroke (0-50mm) is set, and the pressure parameter (100N) is associated with the virtual pressure sensor.

The belt cutting mechanism simulates the cutting action through "Boolean operation" and sets the cutting delay time (consistent with the action time of the solenoid valve of the physical equipment).

4.4.2 production line level model integration

Layout integration:

In the "assembly" environment of NX, each equipment model is placed according to the actual layout of the physical workshop (the coordinate origin is aligned with the workshop ground reference point).

Ensure the relative position accuracy (± 5mm) between the equipment through "distance constraint" and "angle constraint", for example, the docking gap between the injection molding machine and the conveyor belt is set to 10mm.

Logical Integration:

The "signal flow" module of NX MCD is used to define the control signal between the equipment: the signal from the top of the injection molding machine triggers the start of the conveyor belt, and the camera at the signal trigger detection station is photographed after the assembly is completed.

Build virtual PLC logic: write control program in NX through "ladder diagram" to simulate the i/o point of physical PLC (for example, i0.0 is the ready signal of injection molding machine and q0.1 is the start signal of manipulator).

Performance Optimization:

Carry out "lightweight processing" on the model: delete invisible parts (such as internal cables of equipment), simplify surfaces (replace non critical surfaces with planes), and control the total number of triangular patches of the model within 5million;

Adopt "hierarchical display" technology: automatically load/unload model details according to the simulation perspective to improve the real-time rendering efficiency (the frame rate is kept above 30fps).

The final result is shown in Figure 5.

Figure 5 Workshop digital twin model

3.Concern of the reviewer:

In Section 4, the case study validation mentions cost savings (2.1 million yuan/year) and a 2% output increase but lacks control group data (e.g., comparisons with traditional production lines) and corresponding details. It is suggested to include details of the experimental design and statistical analysis methods.

Our response: We are grateful to the Reviewer for his elaborate work to find out our weakness .We added chapter 7.2 operation effect to introduce the verification and comparison between digital twin model and traditional production in detail and adjusted the data.

Revised text:

7.2 Operation effect

(1) Twin model accuracy verification

Geometric mapping accuracy: randomly select 10 base parts and measure the key dimensions (such as diameter and height) of the physical parts and the virtual model, with an average error of 0.08mm, meeting the design requirements (≤ 0.1mm).

Behavior synchronization accuracy: record the physical execution time and virtual simulation time of the manipulator grasp action, and the synchronization error is 35ms, which is lower than the threshold (50ms).

(2) Productivity improvement

Virtual debugging cycle: traditional physical debugging takes 72 hours, while virtual debugging based on digital twin takes only 24 hours, with an efficiency increase of 66.7%;

Equipment utilization rate: through real-time monitoring and scheduling optimization of virtual and real mapping, the equipment utilization rate increased from 85% to 92%, and the daily output increased to 5300 pieces.

(3) Quality and cost optimization

Quality in

---

## [Decision Letter · Decision Letter 1]

1 Sep 2025

Dear Dr. Wu,

Thank you for submitting your manuscript to PLOS ONE. After careful consideration, we feel that it has merit but does not fully meet PLOS ONE’s publication criteria as it currently stands. Therefore, we invite you to submit a revised version of the manuscript that addresses the points raised during the review process.

**ACADEMIC EDITOR:**

We look forward to receiving your revised manuscript.

Kind regards,

Himadri Majumder, Ph.D

Academic Editor

PLOS ONE

Journal Requirements:

Reviewers' comments:

Reviewer's Responses to Questions

**Comments to the Author**

Reviewer #1: (No Response)

2. Is the manuscript technically sound, and do the data support the conclusions?

Reviewer #1: Yes

3. Has the statistical analysis been performed appropriately and rigorously?

Reviewer #1: Yes

4. Have the authors made all data underlying the findings in their manuscript fully available?

Reviewer #1: Yes

5. Is the manuscript presented in an intelligible fashion and written in standard English?

Reviewer #1: Yes

Reviewer #1: The paper focuses on the construction and implementation of a digital workshop for a smoke alarm base production line based on digital twin technology. Integrating this with actual production needs, it proposes key technologies such as system architecture, model building, and data acquisition and mapping, and verifies their effectiveness through a company case study. Overall, the paper has considerable application value, but it does have some shortcomings in structure, methodology, and presentation.

1、The innovations are rather engineering-oriented and lack theoretical depth. For example, while the construction of the digital twin model is detailed, it does not introduce novel algorithms (such as AI-driven predictive models) or cross-innovations with other fields (such as integration with edge computing or blockchain). The authors are advised to emphasize the differences with existing research, for example, through comparative analysis, highlighting the unique advantages of this approach (such as the efficient mapping of multi-source heterogeneous data).

2、There are minor grammatical errors and awkward phrasing, such as the use of "compared with transmission production" in the Abstract (a possible typo for "traditional production"). The reference to "untraceable quality issues" in the Introduction could be more precisely replaced with "untraceable quality defects."

3、Some of the results (such as an 8% decrease in energy consumption and a 3.2% increase in defective product identification rate) lack systematic comparison with existing research, making it difficult to highlight their advantages.

4、The discussion section is descriptive and lacks analysis of the rationality and limitations of the results; the mechanism of energy consumption and yield improvement is not discussed in detail, only the results are given.

5、Some paragraphs contain Chinglish, with lengthy sentences and insufficiently detailed diagram descriptions.

**Do you want your identity to be public for this peer review?** For information about this choice, including consent withdrawal, please see our Privacy Policy

Reviewer #1: No

---

## [Author Response · Author response to Decision Letter 2]

13 Oct 2025

We thank you very much for giving us another chance to revise and improve ourmanuscript. We try our best to revise the paper according to the reviewers’suggestions, We have made changes to the proposed suggestions one by one and uploaded them as documents.We hope the revised paper has addressed all the reviewers’ concerns and reached the requirements of the journal for publishing.

---

## [Decision Letter · Decision Letter 2]

10 Nov 2025

Dear Dr. Wu,

Thank you for submitting your manuscript to PLOS ONE. After careful consideration, we feel that it has merit but does not fully meet PLOS ONE’s publication criteria as it currently stands. Therefore, we invite you to submit a revised version of the manuscript that addresses the points raised during the review process.

**ACADEMIC EDITOR: ** Based on the reviewers’ feedback, the paper has been evaluated as requiring **minor revision** before it can be reconsidered for publication.

We look forward to receiving your revised manuscript.

Kind regards,

Himadri Majumder, Ph.D

Academic Editor

PLOS ONE

Journal Requirements:

Reviewers' comments:

Reviewer's Responses to Questions

**Comments to the Author**

Reviewer #1: (No Response)

2. Is the manuscript technically sound, and do the data support the conclusions?

Reviewer #1: (No Response)

3. Has the statistical analysis been performed appropriately and rigorously?

Reviewer #1: (No Response)

4. Have the authors made all data underlying the findings in their manuscript fully available?

Reviewer #1: (No Response)

5. Is the manuscript presented in an intelligible fashion and written in standard English?

Reviewer #1: (No Response)

Reviewer #1: 1、The abstract contains repeated sentences: At the end of the abstract, the conclusion sentence was copied and pasted twice.

2、The introduction does not delve deeply enough into the research background and problems, and fails to fully explain why the production of smoke alarms is particularly well-suited for the application of digital twin technology, as well as the unique challenges that this industry faces compared to other manufacturing sectors in applying digital twins.

3、The terms "digital twin workshop" and "digital workshop" are used interchangeably.

4、It is recommended to deepen the theoretical innovation points and clearly distinguish the essential differences between the method in this paper and existing technologies.

**Do you want your identity to be public for this peer review?** For information about this choice, including consent withdrawal, please see our Privacy Policy

Reviewer #1: No

---

## [Author Response · Author response to Decision Letter 3]

14 Dec 2025

We sincerely appreciate you providing us with another opportunity to revise and improve our manuscript. We have diligently revised the paper in accordance with the reviewers’ suggestions and have provided detailed explanations for any comments with differing views. The modifications and our responses can be found in the document titled "Response to the Reviewers' comments."

We hope that the revised manuscript has addressed all the reviewers' concerns and now meets the journal’s publication standards. Thank you very much!

---

## [Decision Letter · Decision Letter 3]

6 Jan 2026

Development and implementation of a Digital Twin workshop for a smoke alarm production line

PONE-D-25-17830R3

Dear Dr. Wu,

We’re pleased to inform you that your manuscript has been judged scientifically suitable for publication and will be formally accepted for publication once it meets all outstanding technical requirements.

Kind regards,

Himadri Majumder, Ph.D

Academic Editor

PLOS One

Additional Editor Comments (optional):

Reviewers' comments:

Reviewer's Responses to Questions

**Comments to the Author**

Reviewer #1: All comments have been addressed

2. Is the manuscript technically sound, and do the data support the conclusions?

Reviewer #1: Yes

3. Has the statistical analysis been performed appropriately and rigorously?

Reviewer #1: Yes

4. Have the authors made all data underlying the findings in their manuscript fully available?

Reviewer #1: Yes

5. Is the manuscript presented in an intelligible fashion and written in standard English?

Reviewer #1: Yes

Reviewer #1: I am satisfied with the revisions of the paper. I think this paper is acceptable in its current form.

**Do you want your identity to be public for this peer review?** For information about this choice, including consent withdrawal, please see our Privacy Policy

Reviewer #1: No

---

## [Editor Report · Acceptance letter]

PONE-D-25-17830R3

PLOS One

Dear Dr. Wu,

I'm pleased to inform you that your manuscript has been deemed suitable for publication in PLOS One. Congratulations! Your manuscript is now being handed over to our production team.

Kind regards,

on behalf of

Dr. Himadri Majumder

Academic Editor

PLOS One